# Towards Interpretable and Efficient Attention: Compressing All by Contracting a Few

**Qishuai Wen, Zhiyuan Huang, and Chun-Guang Li**
School of Artificial Intelligence,
Beijing University of Posts and Telecommunications, Beijing 100876, P.R. China
`{wqs, huangzhiyuan, lichunguang}@bupt.edu.cn`

## Abstract

Attention mechanisms have achieved significant empirical success in multiple fields, but their underlying optimization objectives remain unclear yet. Moreover, the quadratic complexity of self-attention has become increasingly prohibitive. Although interpretability and efficiency are two mutually reinforcing pursuits, prior work typically investigates them separately. In this paper, we propose a unified optimization objective that derives inherently interpretable and efficient attention mechanisms through algorithm unrolling. Precisely, we construct a gradient step of the proposed objective with a set of forward-pass operations of our *Contract-and-Broadcast Self-Attention* (CBSA), which compresses input tokens towards low-dimensional structures by contracting a few representatives of them. This novel mechanism can not only scale linearly by fixing the number of representatives, but also covers the instantiations of varied attention mechanisms when using different sets of representatives. We conduct extensive experiments to demonstrate comparable performance and superior advantages over black-box attention mechanisms on visual tasks. Our work sheds light on the integration of interpretability and efficiency, as well as the unified formula of attention mechanisms. Code is available at this https URL.

## 1 Introduction

Attention mechanisms have been widely applied across diverse areas, including computer vision [1, 2], natural language processing [3, 4], and scientific discovery [5]. Nonetheless, a series of puzzling phenomena—such as emergent segmentation properties [6], in-context learning ability [7], attention collapse [8, 9] and extreme-token phenomena [10]—have been uncovered in them, hindering the principled and trustworthy development. At the same time, the quadratic computational and memory complexity of self-attention with respect to the sequence length impedes its broader applications in real-time systems [11], as well as the processing of long documents [12] and high-resolution images [13].

In light of these challenges, it has been more crucial to mathematically demystify attention mechanisms, which offers deeper insights into their simplification and acceleration. Over the past few years, remarkable advances have been made in addressing the interpretability or efficiency issue separately. On the one hand, in an ante-hoc manner, attention mechanisms can be interpreted by optimization objectives grounded in clustering [14], denoising [15], energy minimization [16], matrix decomposition [17], and contrastive learning [18]. These inherently interpretable approaches are more rigorous than post-hoc explanations [19]. On the other hand, numerous techniques have been developed to alleviate the quadratic complexity of self-attention, including sparse attention [20] and linear attention [21].

However, the joint development of interpretability and efficiency in attention mechanisms remains a largely unexplored area of research. This leaves the design of efficient attention mostly heuristic, and the interpretations and explanations for attention mechanisms less instructive. To bridge this gap, we formulate a unified optimization objective by mildly modifying a compression-driven optimization objective called $MCR^2$ [22]. Indeed, this objective has been utilized for designing an interpretable softmax attention, MSSA [23], and a linear-time attention, TSSA [24]. But MSSA also scales quadratically, and TSSA is effectively a channel attention mechanism, which contrasts sharply with both softmax attention (token mixer) and linear attention (channel mixer).[1] Therefore, instead of an isolated mechanism, we aim to develop a framework that unifies these varied attention mechanisms in an interpretable way, revealing how they are fundamentally connected yet distinctly presented, as well as the trade-off between expressive capacity and efficiency.

In this paper, we adopt two ante-hoc interpretations to constitute our proposed optimization objective: a) input tokens are compressed towards low-dimensional structures for compact and structured representation; and b) the geometry and information-theoretic essence of input tokens can be captured by a small number of representatives [25, 26] of them. Since the former has been formulated as the $MCR^2$ objective [22, 23] (see Section 2), the remaining task is to leverage the representatives to optimize it, thereby efficiently compressing all by contracting a few (see Section 3.1). By unrolling the resulting optimization objective, we derive our *Contract-and-Broadcast Self-Attention* (CBSA), which contracts the representatives and broadcasts the contractions back to input tokens (see Section 3.2).

Given a fixed number of representatives, the computational and memory complexity of CBSA scales linearly with the number of input tokens. Moreover, CBSA covers the instantiations of varied attention mechanisms, including softmax attention, linear attention, and channel attention, by taking different sets of representatives (see Section 3.3). As a result, CBSA serves as a unified formula for these attention mechanisms, and attributes their differences to their distinct information propagation (more precisely, compression) patterns induced by the different number and structure of representatives.

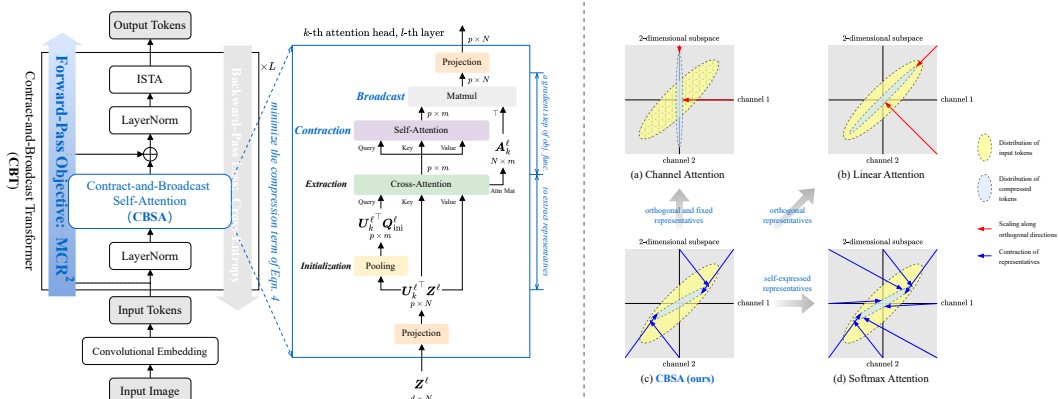

Figure 1: **Overview of CBSA.** *Left panel*: Besides projecting tokens onto subspaces and back the ambient space, there are generally two stages in CBSA: 1) representative initialization and extraction; 2) representative contraction and contraction broadcast. The former extracts representatives satisfying the inequality constraints in (4), while the latter is a gradient step of the compression term in (4). *Right panel*: CBSA covers instantiations of varied attention mechanisms. Their compression patterns are distinct as illustrated above. Further analysis is elaborated in Section 3.3.

**Paper contributions** The contributions of the paper are summarized as follows.

1. We formulate an optimization objective that unifies the interpretability and efficiency of attention mechanisms through the idea of *compressing all by contracting a few*.
2. We derive an inherently interpretable and efficient attention mechanism, CBSA, which is a potential unified formula for different attention mechanisms.
3. We validate the interpretability and efficiency of CBSA through extensive experiments on visual tasks.

---

[1]Softmax attention calculates all possible pairwise similarities between tokens, while TSSA just scales feature channels according to their second moments; see (13).

## 2 Notations and preliminaries

**Notations.** Given a positive integer $n$, let $[n] \doteq \{1, 2, \ldots, n\}$. For $s \geq n$, let $\mathcal{O}(s, n) \subseteq \mathbb{R}^{s \times n}$ denote the set of $s \times n$ matrices with orthonormal columns, and $\mathcal{O}(s) \doteq \mathcal{O}(s, s)$ denote the set of $s \times s$ orthogonal matrices. Let $\mathbf{I}_n$ denote an identity matrix of size $n$, and $\mathbf{O}_n$ denote a zero square matrix of size $n$. Given a vector $\boldsymbol{v} \in \mathbb{R}^n$, let $\mathrm{Diag}(\boldsymbol{v}) \in \mathbb{R}^{n \times n}$ be a diagonal matrix with the entries of $\boldsymbol{v}$ along its diagonal. Let $\boldsymbol{Z} \in \mathbb{R}^{d \times N}$ denote $N$ input tokens represented in the ambient space $\mathbb{R}^d$. Specially, let $\boldsymbol{Z}^\ell$ denote these tokens feeding into the $\ell$-th attention layer. The same holds for the representatives of input tokens, $\boldsymbol{Q} \in \mathbb{R}^{d \times m}$, where $m$ can be much smaller than $N$.

**Union of subspaces.** Although the union of nonlinear manifolds provides a better approximation [27], we adopt a much simpler structure: the union of (low-dimensional) linear subspaces.[2] Specifically, it is parameterized as $K$ incoherent $p$-dimensional subspaces spanned by orthonormal bases $\boldsymbol{U}_{[K]} \doteq \{\boldsymbol{U}_k \in \mathcal{O}(d, p)\}_{k=1}^K$, where $pK = d$ and $\boldsymbol{U}_i^\top \boldsymbol{U}_j = \mathbf{O}_p, \forall i \neq j$. We refer to the $i$-th basis vector of $\boldsymbol{U}_k$ as $\boldsymbol{u}_{ki}$. Similar to [23], we implement $\boldsymbol{U}_{[K]}$ as learnable parameters in each attention layer and thus denote the $\boldsymbol{U}_{[K]}$ implemented by the $\ell$-th layer as $\boldsymbol{U}_{[K]}^\ell = \{\boldsymbol{U}_k^\ell\}_{k=1}^K$.

**Coding rate.** To quantify the compactness, i.e., the extent to which tokens are compressed towards subspaces, we adopt the (lossy) *coding rate* [28], which measures how efficiently the token distribution can be covered by $\epsilon$-balls under a given quantization precision $\epsilon > 0$ (as illustrated in Fig. 1(a)). The coding rate of input tokens in the ambient space $\mathbb{R}^d$ is defined as:

$$R(\boldsymbol{Z}) \doteq \frac{1}{2} \log \det \left( \mathbf{I}_N + \frac{d}{N\epsilon^2} \boldsymbol{Z}^\top \boldsymbol{Z} \right). \tag{1}$$

**MCR$^2$ objective.** The Maximal Coding Rate Reduction (MCR$^2$) [22] objective adopted in [23] is defined on the coding rate as follows:

$$\max_{\boldsymbol{Z}} \ \Delta R(\boldsymbol{Z}) \doteq \underbrace{R(\boldsymbol{Z})}_{\text{expansion}} - \underbrace{R_c(\boldsymbol{Z} \mid \boldsymbol{U}_{[K]})}_{\text{compression}} - \underbrace{\lambda \|\boldsymbol{Z}\|_0}_{\text{sparsity}} \doteq R(\boldsymbol{Z}) - \sum_{k=1}^K R(\boldsymbol{U}_k^\top \boldsymbol{Z}) - \lambda \|\boldsymbol{Z}\|_0. \tag{2}$$

The input tokens are compressed towards $K$ subspaces by the compression term, while expanded in the ambient space by the expansion term to avoid collapse, yielding compact and structured representation [22]. Yu et al. [23] have demonstrated that the approximated gradient step of the compression term in (2) corresponds an interpretable softmax attention mechanism.

## 3 Methods

### 3.1 Compressing all by contracting a few

Due to the existence of Gram matrix in (1), the attention mechanism derived from (2), which is called Multi-head Subspace Self-Attention (MSSA), inevitably scales quadratically with the number of input tokens. While previous work has bypassed this issue by replacing the Gram matrix with the covariance matrix [29] and further introduced a variational formulation [24], these strategies degenerate the token mixer into a channel mixer and channel attention, respectively.

In this paper, inspired by the concept of landmarks [30, 31], we propose a simple but flexible approach to streamline the optimization of MCR$^2$: *compressing all input tokens by contracting a small number of representatives of them.* Before demonstrating that this achieves linear complexity in $N$ and prevents the aforementioned degeneration, we first formulate it as a new optimization objective.

An initial attempt is to impose a set of equality constraints on the coding rates as follows:

$$\max_{\boldsymbol{Z}} \ R(\boldsymbol{Z}) - \sum_{k=1}^K R(\boldsymbol{U}_k^\top \boldsymbol{Q}) - \lambda \|\boldsymbol{Z}\|_0 \ \text{s.t.} \ R(\boldsymbol{U}_k^\top \boldsymbol{Q}) = R(\boldsymbol{U}_k^\top \boldsymbol{Z}), \ \forall k \in [K], \tag{3}$$

---

[2]As this structure faces challenges in adapting to all modalities and tasks, we excluded natural language processing tasks from our experiments. But we argue that it serves as a feasible starting point for finer-grained structures; further discussion is available on our OpenReview page.

where representatives $\boldsymbol{Q} \doteq q(\boldsymbol{Z})$ are extracted from input tokens $\boldsymbol{Z}$ by a differential function $q(\cdot) : \mathbb{R}^{d \times N} \to \mathbb{R}^{d \times m}$. [3] This new objective in (3) is equivalent to the original objective in (2) but is more efficient to handle because the number of representatives (e.g., $m = p = d/\kappa$) can be far smaller.

Since that the equality constraints in (3) is overly restrictive in practice, we attempt to relax them by introducing a tolerance $\tau$, which uniformly bounds the absolute difference of the two coding rates within each subspace, i.e., $|R(\boldsymbol{U}_k^\top \boldsymbol{Q}) - R(\boldsymbol{U}_k^\top \boldsymbol{Z})| \leq \tau$. Therefore, contracting the representatives will correspondingly compress the input tokens as well up to the tolerance $\tau$.

Consequently, we have a relaxed optimization problem for our subsequent derivations as follows:

$$\max_{\boldsymbol{Z}} \ R(\boldsymbol{Z}) - \sum_{k=1}^{K} R(\boldsymbol{U}_k^\top \boldsymbol{Q}) - \lambda \|\boldsymbol{Z}\|_0 \ \text{ s.t. } \ |R(\boldsymbol{U}_k^\top \boldsymbol{Q}) - R(\boldsymbol{U}_k^\top \boldsymbol{Z})| \leq \tau, \ \forall k \in [K]. \quad (4)$$

In this paper, we employ an arguably simplest way to extract $\boldsymbol{Q}$ in each subspace: $\boldsymbol{U}_k^\top \boldsymbol{Q} = \boldsymbol{U}_k^\top \boldsymbol{Z} \boldsymbol{A}_k$, where $\boldsymbol{A}_k \in \mathbb{R}^{N \times m}$ is the coefficient matrix over dictionary $\boldsymbol{U}_k^\top \boldsymbol{Z} \in \mathbb{R}^{p \times N}$, i.e., the projected representatives in each subspace are linear combinations of the projected input tokens.

### 3.2 Contract-and-Broadcast Self-Attention

We now are ready to derive an attention mechanism in an ante-hoc interpretable manner, by implementing a gradient step of the compression term in the proposed objective as its forward pass. This methodology dates back its origins to the pioneering work [32], and is referred to as algorithm unrolling or unfolding [33].

**Representative initialization and extraction.** Inspired by the fact that cross-attention can be interpreted as approximating the coding rate [34], we employ this idea to extract representatives that satisfy the inequality constraints in (4), thus capturing the information-theoretic essence of input tokens. Specifically, we take an initial guess of the representatives as the query, and the input tokens as the key and value matrices, i.e.,

$$\boldsymbol{U}_k^\top \boldsymbol{Q} = \boldsymbol{U}_k^\top \boldsymbol{Z} \underbrace{\text{softmax}\left((\boldsymbol{U}_k^\top \boldsymbol{Z})^\top (\boldsymbol{U}_k^\top \boldsymbol{Q}_{\text{ini}})\right)}_{\boldsymbol{A}_k}, \ \forall k \in [K], \quad (5)$$

where the initial guess $\boldsymbol{Q}_{\text{ini}}$ is treated as a constant with respect to $\boldsymbol{Z}$ such that its strategy (whether input-dependent or not) does not affect the subsequent derivation. To be more specific, following [35], we initialize $\boldsymbol{Q}_{\text{ini}}$ via an average pooling over the input tokens; see Appendix A for discussion. More importantly, the representatives extracted by this cross-attention operation are linear combinations of the input tokens, thus naturally leading to the form we desire. Therefore, the attention matrix in (5) effectively is the coefficient matrix $\boldsymbol{A}_k$.

**Representative contraction and contraction broadcast.** To derive the attention mechanism, following [23], we focus on optimizing the compression term via a gradient descent step. Having the extracted representatives satisfying the inequality constraints, we take a gradient descent step on the compression term of the objective function in (4) with respect to input tokens as follows:

$$\boldsymbol{Z} \leftarrow \boldsymbol{Z} - \kappa \, \text{CBSA}(\boldsymbol{Z} \mid \boldsymbol{U}_{[K]}) \ \text{ where} \quad (6)$$

$$\text{CBSA}(\boldsymbol{Z} \mid \boldsymbol{U}_{[K]}) \doteq \sum_{k=1}^{K} \boldsymbol{U}_k \underbrace{\boldsymbol{U}_k^\top \boldsymbol{Q} \left(\mathbf{I}_m + \frac{p}{m\epsilon^2}(\boldsymbol{U}_k^\top \boldsymbol{Q})^\top (\boldsymbol{U}_k^\top \boldsymbol{Q})\right)^{-1}}_{\text{Contraction}} \underbrace{\boldsymbol{A}_k^\top}_{\text{Broadcast}}, \quad (7)$$

in which the step size parameter $\kappa$ is learnable in our implementation. One can verify that (7) is proportional to the gradient of the compression term in (4) with respect to input tokens, while the contraction term in (7) is proportional to the gradient with respect to representatives. Hence, we refer to this formula as a *Contract-and-Broadcast Self-Attention* (CBSA), reflecting that: a) the contraction term gives the contracting directions of the representatives (abbreviated as contractions);

---

[3]The representatives are not necessarily a subset of the input tokens; they resemble the cluster centroids in $k$-means, which are continuously extracted, rather than discretely selected.

b) the broadcast term, which reuses the attention matrix in (5), broadcasts the contractions back to all input tokens.

**Contraction via self-attention.** To avoid computing the expensive matrix inverse in (7), similar to [23], we approximate it by a Gram matrix and a softmax function (i.e., an attention matrix):[4]

$$\text{CBSA}(\boldsymbol{Z} \mid \boldsymbol{U}_{[K]}) \approx \underbrace{\boldsymbol{U}_k^\top \boldsymbol{Q} \, \text{softmax}\left((\boldsymbol{U}_k^\top \boldsymbol{Q})^\top(\boldsymbol{U}_k^\top \boldsymbol{Q})\right)}_{\text{Contraction via self-attention}} \underbrace{\boldsymbol{A}_k^\top}_{\text{Broadcast}} . \tag{8}$$

Note that the contraction term now effectively constitutes a self-attention operation in which the linear projections for the query, key, and value are all identical to the subspace basis, i.e., $\boldsymbol{W}_{\text{query}} = \boldsymbol{W}_{\text{key}} = \boldsymbol{W}_{\text{value}} = \boldsymbol{U}_k^\top$. By default, we implement CBSA via (8) rather than (7) in our experiments.

**Overview of CBSA.** The workflow of CBSA is illustrated in the left panel of Fig. 1. We also construct an inherently interpretable *Contract-and-Broadcast Transformer* (CBT) by stacking CBSA with the ISTA module [23], which is also derived via algorithm unrolling.[5] We report the computational complexities of CBSA and its sub-operations in Table 1 and compare the FLOPs of different attention mechanisms in Fig. 2 where $d = 384$, $H = 6$, and a patch size of $16 \times 16$. Provided $N > 2d/H = 2p$ (typically 128 in Transformers), the FLOPs of CBSA are lower than those of MSSA. Further comparisons with other modules, including MHSA and MLP, are provided in Fig. 9.

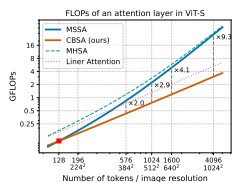

Figure 2: Computation complexity.

Table 1: **Computational complexities.** By default, we set $m = p = d/H$, where $H = K$ denotes the number of attention heads (interpreted as subspaces in our case). The complexity of each sub-operation is computed by summing the costs across all heads. It is worth noting that the projection operations, which are essential to almost all attention mechanisms, confine the overall complexity at least $\mathcal{O}(Nd^2)$.

| $\Omega(\text{MSSA})$ | $\Omega(\text{CBSA})$ | sub-operations of CBSA | | |
| | | $\Omega(\text{extraction}) / \Omega(\text{broadcast})$ | $\Omega(\text{contraction})$ | $\Omega(\text{projection})$ |
| --- | --- | --- | --- | --- |
| $2Nd^2 + 2N^2d$ | $2Nd^2 + 3Nmd + 2m^2d$ | $Nmd$ | $m^2d$ | $Nd^2$ |

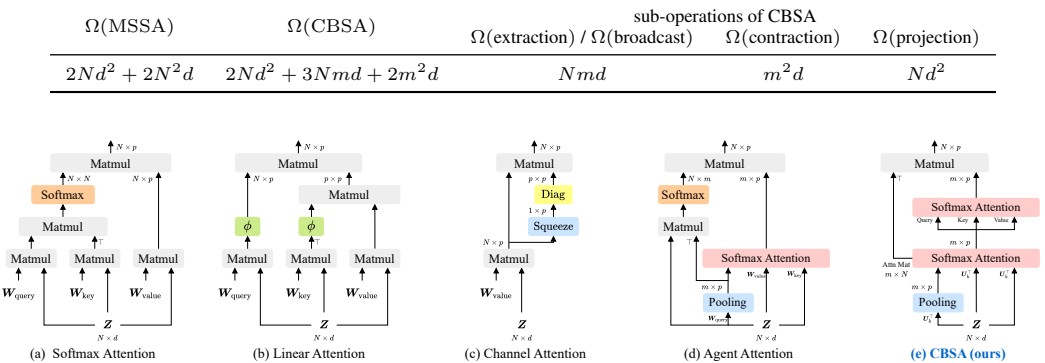

Figure 3: **Different attention mechanisms.** The mapping function and agent bias in the diagram of Agent Attention [35] are omitted for simplicity, and $\top$ stands for the matrix transpose.

## 3.3 CBSA as a unified attention formula

In this subsection, we explore the potential of CBSA to serve as a unified formula for different attention mechanisms. Unlike recent work [37, 38], CBSA encompasses a broader spectrum of mechanisms (see Fig. 3) in an interpretable and mathematically grounded manner.

Our analysis reveals that, CBSA (7) can derive multiple variants corresponding to existing attention mechanisms by varying the choice of representatives. In these variants, the distinct initialization, extraction, contraction, and broadcast steps of CBSA may not be explicitly observed, as some of them are simplified or fused together due to the specific number and structure of representatives, or engineering concerns.

**Softmax attention variant.** Obviously, the input tokens themselves satisfy the constraints in (4), and thus can be directly used as the representatives, i.e., $\boldsymbol{Q} = \boldsymbol{Z}$ and $m = N$. In this case, we call the

---

[4]The scaling factor and sign inversion [36] introduced by this approximation are absorbed into the learnable step size $\kappa$ for notational simplicity.

[5]Note that the ISTA module is designed to unroll the suitably-relaxed proximal gradient step to address the difference of the sparsity penalty and the expansion term $\lambda\|\boldsymbol{Z}\|_0 - R(\boldsymbol{Z})$.

input tokens are *self-expressed* [39], where data samples are linearly represented over a dictionary composed of themselves. Then we can take a trivial solution for the regularization term where all coefficient matrices are identity matrices. Substituting them into (8) yields the following operator, known as MSSA in the white-box transformer [23]:

$$\mathrm{MSSA}(\boldsymbol{Z} \mid \boldsymbol{U}_{[K]}) \doteq \sum_{k=1}^{K} \boldsymbol{U}_k \underbrace{\boldsymbol{U}_k^\top \boldsymbol{Z}}_{\text{v.s. } \boldsymbol{W}_{\text{value}} \boldsymbol{Z}} \underbrace{\mathrm{softmax}\left((\boldsymbol{U}_k^\top \boldsymbol{Z})^\top (\boldsymbol{U}_k^\top \boldsymbol{Z})\right)}_{\text{v.s. } \mathrm{softmax}((\boldsymbol{W}_{\text{key}} \boldsymbol{Z})^\top (\boldsymbol{W}_{\text{query}} \boldsymbol{Z}))}. \qquad (9)$$

**Linear attention variant.** To analyze the case of orthogonal representatives, we start with a canonical choice: the principal directions of input tokens. We thus perform singular value decomposition (SVD) within each subspace:

$$\boldsymbol{U}_k^\top \boldsymbol{Z} = \boldsymbol{L}_k \boldsymbol{\Sigma}_k \boldsymbol{R}_k^\top, \forall\, k \in [K], \qquad (10)$$

where $\boldsymbol{L}_k \in \mathcal{O}(p)$, $\boldsymbol{R}_k \in \mathcal{O}(N, p)$, $\boldsymbol{\Sigma}_k$ is a $p \times p$ diagonal matrix of singular values, and the columns of $\boldsymbol{L}_k$ are known as the principal directions. Then, by right multiplying both sides by $\boldsymbol{R}_k$, we have:

$$\boldsymbol{U}_k^\top \boldsymbol{Z} \boldsymbol{R}_k = \boldsymbol{L}_k \boldsymbol{\Sigma}_k \boldsymbol{R}_k^\top \boldsymbol{R}_k = \boldsymbol{L}_k \boldsymbol{\Sigma}_k, \forall\, k \in [K]. \qquad (11)$$

By comparing (11) with the way to form the representatives, i.e., $\boldsymbol{U}_k^\top \boldsymbol{Q} = \boldsymbol{U}_k^\top \boldsymbol{Z} \boldsymbol{A}_k$, we let $\boldsymbol{U}_k^\top \boldsymbol{Q} = \boldsymbol{L}_k \boldsymbol{\Sigma}_k$ and $\boldsymbol{A}_k = \boldsymbol{R}_k$. Substituting them into (7) leads to the following operator:

$$\sum_{k=1}^{K} \boldsymbol{U}_k \underbrace{\mathcal{F}\left((\boldsymbol{U}_k^\top \boldsymbol{Z})(\boldsymbol{U}_k^\top \boldsymbol{Z})^\top\right)}_{\text{v.s. } \boldsymbol{W}_{\text{value}} \boldsymbol{Z} \phi(\boldsymbol{W}_{\text{key}} \boldsymbol{Z})^\top} \underbrace{\boldsymbol{U}_k^\top \boldsymbol{Z}}_{\text{v.s. } \phi(\boldsymbol{W}_{\text{query}} \boldsymbol{Z})} \doteq \sum_{k=1}^{K} \boldsymbol{U}_k \boldsymbol{L}_k \left(\mathbf{I}_m + \frac{1}{\epsilon^2} \boldsymbol{\Sigma}_k^2\right)^{-1} \boldsymbol{L}_k^\top \boldsymbol{U}_k^\top \boldsymbol{Z}, \qquad (12)$$

where $\mathcal{F}$ is a function defined on the spectrum of a positive semi-definite matrix and applies $f(\lambda_i) = \epsilon^2/(\epsilon^2 + \lambda_i)$ to each eigenvalues $\{\lambda_i\}_{i=1}^{p}$ of the covariance matrix.[6] This operator highly resembles the linear attention [21], due to that it also factorizes the $N \times N$ attention matrix and multiplies the key and value first to linearize the computational complexity.[7] In Appendix B, we prove that a similar result holds for any set of orthogonal representatives.

**Channel attention variant.** Assuming that the basis vectors of $\boldsymbol{U}_k$ are the principal directions for any set of input tokens (which is *impossible* but simplifies the computation), the directions of the representatives can be fixed along these basis vectors, i.e., $\boldsymbol{U}_k = \boldsymbol{L}_k$, thereby being orthogonal and input-agnostic (fixed). Then, (12) is simplified to:

$$\sum_{k=1}^{K} \boldsymbol{U}_k \boldsymbol{D}_k \boldsymbol{U}_k^\top \boldsymbol{Z}, \text{ where } \boldsymbol{D}_k \doteq \mathrm{Diag}\left([f((\boldsymbol{u}_{ki}^\top \boldsymbol{Z})(\boldsymbol{u}_{ki}^\top \boldsymbol{Z})^\top)]_{i=1}^{p}\right), \qquad (13)$$

which basically recovers TSSA [24]. In (13), the feature channels are adaptively scaled according to their second moments of token projections, while channel attention typical employs an MLP to predict the channel-wise scaling factors [40, 41].

**Agent attention variant.** Agent Attention is basically a variant of CBSA with the contraction step removed[8] which can be perceived in Fig. 3. Although this removal appears to confine the token-mixing ability, it is compensated by the pooling-based initialization, which is also a token mixer [42].

To gain some intuition of the gap in expressive capacity among the aforementioned mechanisms, we illustrate their compression patterns in the right panel of Fig. 1. The channel attention variant is restricted to compressing input tokens along fixed axes parameterized by $\boldsymbol{U}_{[K]}$, whereas the linear attention variant compresses them along principal directions that are dynamically determined by the input. We argue that such a dynamism is crucial for in-context learning [7] and for mitigating

---

[6]As $f$ is monotonically decreasing, the effect of (12) is to preserve the principle directions (i.e., representatives) with large variance while suppressing the other directions with vanishing variance [29].

[7]In fact, (12) is less expressive than black-box linear attention, because $(\boldsymbol{U}_k^\top \boldsymbol{Z})(\boldsymbol{U}_k^\top \boldsymbol{Z})^\top$ is symmetric but $(\boldsymbol{W}_{\text{value}} \boldsymbol{Z})(\boldsymbol{W}_{\text{key}} \boldsymbol{Z})^\top$ is generally not.

[8]Since the query, key, and value are identical in CBSA, the broadcast term can be obtained directly from the extraction step, eliminating the separate computation branch in Agent Attention.

superposition [43]. In contrast, softmax attention exhibits much greater flexibility, as it manipulates each token independently. Actually, its compression can be viewed as operating in an $N$-dimensional space, rather than in the $d$-dimensional feature space. Our proposed CBSA aims to approximate the behavior of softmax attention while significantly reducing computational cost.

The above findings can also be interpreted from a dictionary learning perspective, where the representatives correspond to the atoms of a dictionary. When the representatives are orthogonal and fixed, they form a complete dictionary; when they are orthogonal yet input-dependent, they resemble a submatrix of an overcomplete dictionary as in compressed sensing [44]. When the input tokens themselves serve as representatives, they constitute a self-expressive dictionary [39].

## 4 Experiments

In this section, we evaluate the interpretability and the efficiency of the proposed CBSA and the CBTs built upon CBSA. As natural images often lie on low-dimensional subspaces [45, 46], we focus on classical visual tasks such as image classification and semantic segmentation, where higher resolutions generally lead to better accuracy [47, 48].

**Baseline and training configuration.** We compare our CBSA to the vanilla softmax attention [49, 1], and interpretable attention mechanisms based on $MCR^2$, e.g., CRATE [23], ToST [24] and DEPICT [34]. Table 2 summarizes the baselines with brief descriptions. The results in gray are cited directly from the corresponding papers; whereas the others are reproduced under varied settings for fair comparisons. By default, the training configuration follows the baselines, with detailed information provided in Appendix C.

**Implementation detail.** The projection back to the ambient space, which should theoretically be a left multiplication by $U_k$, is over-parameterized with an independently learnable matrix. This strategy is also adopted in MSSA [23] and TSSA [24], and its effect has been analyzed in [36]. In short, although this relaxation compromises the theoretical rigour, it is crucial for achieving better accuracy. In addition, the step size $\kappa$ in (6) is implemented as a learnable parameter without constraining its sign. This allows the model to flexibly choose between compression and decompression. The PyTorch implementation is provided in Appendix D.

Table 2: **Summary of baselines.** Note that these methods are not limited to the tasks listed here, our descriptions only indicate their usages in the experiments of this paper.

| Methods | Attention Mechanism | Complexity | Interpretable | Tasks |
|---|---|---|---|---|
| ViT [1] | MHSA (softmax attention) | quadratic | ✗ | image classification |
| CRATE [23] | MSSA (softmax attention) | quadratic | ✓ | image classification |
| ToST [24] | TSSA (channel attention) | linear | ✓ | image classification |
| Agent Attention [35] | Agent Attention (linear attention) | linear | ✗ | image classification |
| Segmenter [50] | MHSA | quadratic | ✗ | semantic segmentation |
| DEPICT [34] | MSSA | quadratic | ✓ | semantic segmentation |

### 4.1 Advantages enabled by interpretability

In this subsection, we show superior advantages of CBSA over black-box attention mechanisms. The most essential aspect of our CBSA is its interpretability, which induces other desirable properties such as robustness and emergent segmentation.

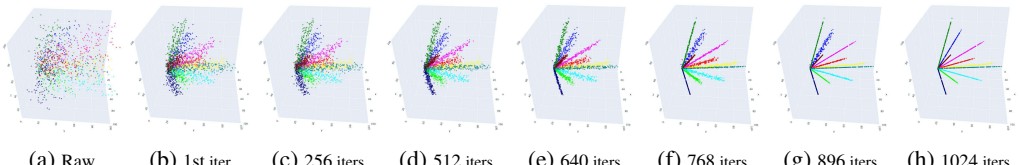

(a) Raw    (b) 1st iter    (c) 256 iters    (d) 512 iters    (e) 640 iters    (f) 768 iters    (g) 896 iters    (h) 1024 iters

Figure 4: **Compact and structured representation.** There are 10 classes indicated by colors. Points in each class are generated by sampling from a one-dimensional subspace and are then perturbed by adding noise.

**Compact and structured representation.** Similar to $MCR^2$ [22], our optimization objective (4) aims to learn compact and structured representation by compressing input tokens towards low-dimensional subspaces. To confirm whether its iterative gradient steps can actually achieve this goal, we iterates

the linear attention variant of CBSA (12) on synthetic data, where image tokens are modeled as the points in a three-dimensional space $\mathbb{R}^3$. Specifically, it is conducted on each class in the ambient space (thus being parameter-free) with a forward-only manner. As shown in Fig. 4, the representation ultimately admits a union of well-separated one-dimensional subspaces after $1024$ iterations.

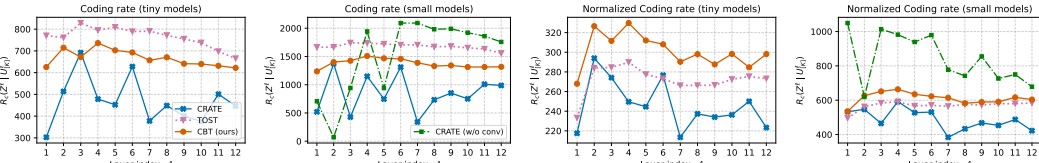

Figure 5: **Evaluation on compression effect.** We measure the compression term of (2) as a function of layers.

**Compressing all by contracting a few.** To confirm our interpretation that CBSA compresses all input tokens by contracting a few representatives, we check that: *whether the input tokens are indeed compressed; if so, whether the compression is driven by contracting the representatives.* We measure the compression term of (2) as well as its normalized variant[9] in Fig. 5. This normalized coding rate is invariant to in-place scaling and depends on the angles between input tokens. We observe two pieces of supporting evidences: a) the more compact the ultimate representation is, i.e., the compression term measured in the last layer is lower, the better the model performs on ImageNet-1K (see Table 3);[10] b) the latter half of the layers exhibit consecutive compression, in nearly all models.[11] Then, in Fig. 6, we measure the reduced coding rate of the input tokens and representatives, respectively, after they are processed by CBSA within each subspace. We observe that the two kinds of reduced coding rates show highly similar trends across most subspaces.[12]

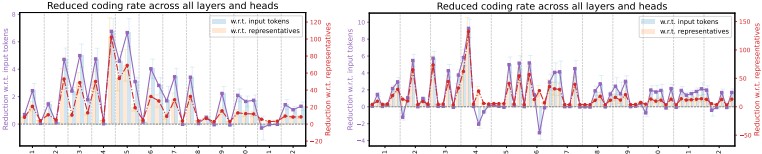

Figure 6: **Comparison on reduced coding rates.** The reduction with respect to input tokens is calculated between input tokens and compressed tokens, while the reduction with respect to representatives is calculated between extracted representatives and contracted representatives. We measure the reduced coding rate with respect to the input tokens and the representatives, respectively, across all heads of a model. Results of CBT-T/S are presented here, those of CBT-B/L can be found in Fig. 10. Note that we set $\kappa = 1$ to exclude the de-compression cases observed in Fig. 5.

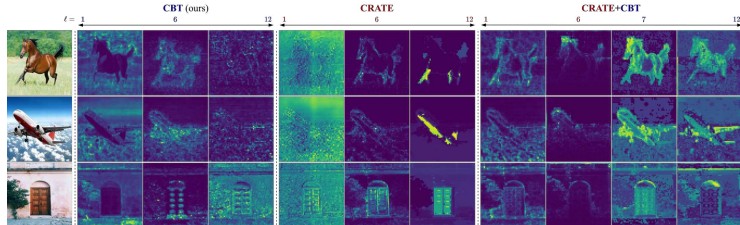

Figure 7: **Visualization of [CLS] attention map.** We empirically estimate the full attention matrix by $\boldsymbol{A}_k \boldsymbol{A}_k^\top \in \mathbb{R}^{N \times N}$ and visualize the attention maps of the [CLS] token from the early, middle, and late layers of CBT, CRATE, and their hybrid model, respectively.

**Emergent segmentation properties.** It has been reported that segmentation properties emerge in CRATE with merely standard supervised classification training owing to MSSA [51]. Compared to CRATE, our CBT attends to more semantically meaningful regions in the early layers, but the segmentation properties fail to persist in subsequent layers. as shown in Fig. 7. To address this

---

[9]That is, the input tokens are normalized to unit vector before measuring their coding rate.

[10]Note that this correlation does not hold for the normalized coding rate, indicating that compression in magnitude is a critical aspect.

[11]We regard the decompression phenomenon in the early layers as a "known unknown" requiring further investigation.

[12]We hypothesis that the higher coding rate of the representatives makes them act as "scalpels" in order to give a "surgery" on the input tokens.

phenomenon, we construct a hybrid model, termed CRATE+CBT, where the first half of the attention layers employ MSSA and the latter half employs CBSA. In this hybrid model, we observed that the segmentation properties not only emerged in the very first layer, but are also progressively enhanced in the following layers, rather than fading as in CBT. Qualitative results supporting this conclusion can be found in Appendix C.

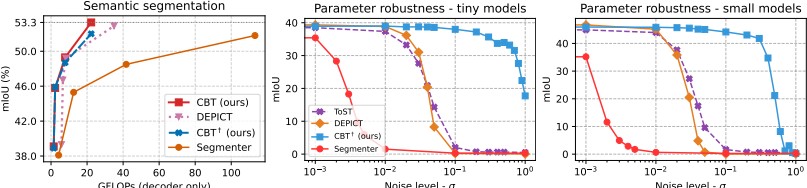

Figure 8: **Semantic segmentation on ADE20K.** All results are evaluated on the ADE20K validation set. CBSA$^†$ indicates that the CBSA layers are implemented rigorously, i.e., without the overparameterization trick. *Middle and Right:* Random Gaussian noises with zero mean and standard deviation $\sigma$ are independently added to each parameter of the attention layers in the decoder.

**Robustness against parameter perturbation.** As the attention heads of CBSA are modeled as low-dimensional subspaces, perturbing their projection matrices (i.e., subspace bases) with relatively small noise does not significantly alter the subspaces they span [34]. Consequently, as shown in the right two panels of Fig. 8, CBSA is extremely robust against parameter perturbation, whereas the black-box method (i.e., Segmenter [50]) collapses under the same perturbation.

Table 3: **Classification on ImageNet-1k.** Models are trained on images of resolution $224 \times 224$ with a patch size of $16 \times 16$ where CBT and ViT use convolutional embedding layers but CRATE uses linear embeddings.

| Datasets | CBT-T(iny) | CBT-S(mall) | CBT-B(ase) | CBT-L(arge) | CRATE-B | CRATE-L | ViT-S |
|---|---|---|---|---|---|---|---|
| # parameters | 1.8M | 6.7M | 25.7M | 83.1M | 22.8M | 77.6M | 22.1M |
| FLOPs | 1.1G | 4.0G | 15.1G | 47.3G | 12.6G | 43.3G | 9.8G |
| ImageNet-1K | 63.2 | 71.4 | 73.4 | 74.4 | 70.8 | 71.3 | 72.4 |
| CIFAR10 | 94.8 | 96.3 | 96.7 | 97.3 | 96.8 | 97.2 | 97.2 |
| CIFAR100 | 76.5 | 80.4 | 82.0 | 83.4 | 82.7 | 83.6 | 83.2 |
| Oxford Flowers-102 | 88.4 | 91.7 | 93.6 | 93.9 | 88.7 | 88.3 | 88.5 |
| Oxford-IIIT-Pets | 86.8 | 91.6 | 92.6 | 92.9 | 85.3 | 87.4 | 88.6 |

Table 4: **Fair comparisons on ImageNet-1K.** All models employ convolutional embedding layers and ISTA feedforward blocks. The only difference lies in the attention mechanism, where Agent-T and Agent-S [35] are implemented as in CBSA but without the contraction step (as derived in Section 3.3).

| ImageNet-1K | CBSA-T | CBSA-S | TSSA-T | TSSA-S | MSSA-T | MSSA-S | Agent-T | Agent-S |
|---|---|---|---|---|---|---|---|---|
| # pairwise similarities | 0.53M | 1.1M | 0.45M | 0.91M | 1.4M | 2.8M | 0.52M | 1.0M |
| Top-1 accuracy | 63.2 | 71.4 | 61.2 | 68.5 | 64.7 | 72.1 | 63.8 | 71.8 |

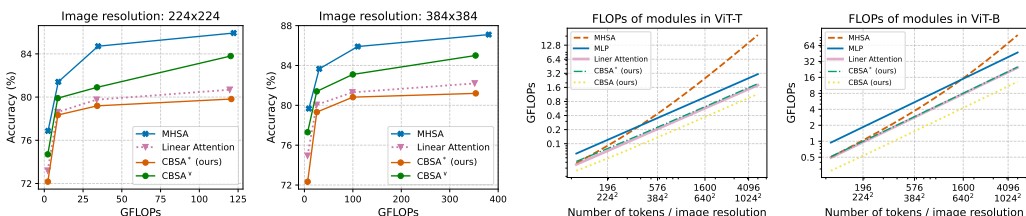

Figure 9: **Adapting pre-trained ViTs into CBSA style.** We finetune both the adapted models and the original ViTs on ImageNet-1k for 50 epochs, and report the top-1 accuracy on the validation set. CBSA* leverages the three distinct projection matrices inherited from the pretrained MHSA to calculate query, key, and value, instead of using a single projection matrix as in (8). CBSA$^∨$ refers to the CBSA* without the contraction step, which is essentially an Agent attention.

## 4.2 Evaluations on real-world visual tasks

We pretrain CBT models on the ImageNet-1k dataset, and finetune them on several downstream datasets. The top-1 accuracy on validation sets is reported in Table 3. In particular, our CBT-Small achieves comparable top-1 accuracy to ViT-S using only 30% of the parameters and 40% of the

FLOPs. Compared to CRATE models, our CBTs (with convolutional embedding layers) perform remarkably better while using fewer parameters and FLOPs.

We also conduct a set of fair comparisons across different attention mechanisms, which can be regarded as variants of CBSA, and report the results in Table 4. In this setting, our CBT models remain competitive with CRATE while computing significantly fewer pairwise similarities. These results confirm that, from MSSA to CBSA and then to TSSA, the number of pairwise similarity computations decreases at the cost of performance sacrifice. In addition, we present the throughput comparisons on high-resolution images (e.g., $512 \times 512$) in Table 7 (Appendix C), showing that our methods consistently achieves superior training and inference efficiency.

To investigate the potential of applying CBSA to large-scale pretraining, we preliminarily finetune ViT models pretrained on ImageNet-21k[13] by adapting their attention blocks into the CBSA style. For comparison, we also adapt them into linear attention. Experimental results are shown in Fig. 9. Although CBSA deviates more from MHSA than linear attention and is consequently harder to adapt from pretrained ViTs, CBSA achieves comparable performance to that of linear attention with nearly the same FLOPs. Interestingly, when the contraction step is removed, CBSA surpasses linear attention, which has been extensively investigated in [35].

For semantic segmentation, following the design of DEPICT [34], we build CBT decoders by stacking CBSA layers without the feed-forward modules on the top of ViT encoders. We evaluate the performance of them on the ADE20K dataset [53] and show the results in the left panel of Fig. 8. Clearly, our CBT decoder consistently surpasses both white-box (DEPICT) and black-box (Segmenter) counterparts that rely on softmax attention. In particular, the best-performing CBT decoder improves upon Segmenter by $1.5\%$ mIoU while using merely $20\%$ of the FLOPs and $0.06\%$ of the pairwise similarities in the decoder.

# 5  Related work

Efficient attention mechanisms can be roughly divided into two categories: sparse attention and linear attention [54]. Approaches in sparse attention sparsify the attention matrix proactively by restricting the attention span to either random, or fixed [55, 20], or learnable [56, 57] patterns, or their combinations. Approaches in linear attention [21, 58] decompose the attention matrix into a product of two low-rank matrices and thus avoids its explicit computation via the associative property of multiplication. The idea of using representative tokens has been applied to both. Global tokens or memory can be introduced in sparse attention to maintain global connectivity, thereby further shrinking the attention span [59, 60]. Meanwhile, Agent tokens [35] and landmarks [30] can also be incorporated into linear attention from different perspectives.

Our CBSA distinguishes itself from previous efficient attention mechanisms by being inherently interpretable and derived from an optimization objective which efficiently compresses input tokens towards low-dimensional structures. Moreover, it can not only be viewed as sparse attention or linear attention for leveraging the representatives, but also mathematically generalizes softmax attention, linear attention, and channel attention as its special cases.

# 6  Conclusion

We have proposed an optimization objective for deriving attention mechanism and unifying the investigation towards the interpretability and the efficiency. By unrolling the gradient optimization steps of this objective, we derived an inherently interpretable and efficient attention mechanism, called Contract-and-Broadcast Self-Attention (CBSA). We found that our CBSA covers the instantiations of softmax attention, linear attention, and channel attention by changing the number and structure of representatives, thus revealing their fundamental connections. We validated the effectiveness of our CBSA through extensive experiments on visual tasks. We believe that the preliminary framework established in this work offers a promising direction for exploring a unified formula for existing attention mechanisms as well as new attention mechanisms in an inherently interpretable way.

---

[13]The checkpoint is obtained from the timm [52] library.

## Acknowledgments and Disclosure of Funding

This work is supported by the National Natural Science Foundation of China under Grant Nos. 62576048 and 61876022. C.-G. Li is the corresponding author.

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

# Appendix

## A  Representative initialization

Besides pooling the input tokens, another intuitive choice for representative initialization is to set them as learnable (i.e., trainable) parameters, similar to the object queries in [61], the class embeddings in [50], and the registers in [62]. However, in this case we found that the attention matrix in the extraction (cross-attention) step[14] is low rank. For example, although its maximal rank can be $m = p = \frac{d}{H} = 64$ in CBT models, it typically remains around 5. This indicates that far fewer initialized representatives are being utilized than expected, leading to unsatisfactory performance of CBSA.

By contrast, when pooling-based initialization is employed, the rank of the attention matrix always attains its maximum, i.e., $m$. Nonetheless, as pointed out in [63], this still constitutes a low-rank bottleneck: linear attention underperforms softmax attention, whose attention matrix has rank $N \gg m$. Although this issue can be alleviated by introducing depth-wise convolution (DWC), which boosts performance while preserving linear complexity, our work does not involve this modification since we focus on theoretical contributions rather than empirical improvements.

## B  Derivations

Here, we show how (12) is derived by substituting $\boldsymbol{U}_k^\top \boldsymbol{Q} = \boldsymbol{L}_k \boldsymbol{\Sigma}_k$, $\boldsymbol{A}_k = \boldsymbol{R}_k$ into (7):

$$\sum_{k=1}^{K} \boldsymbol{U}_k \boldsymbol{L}_k \boldsymbol{\Sigma}_k \left( \mathbf{I}_m + \frac{m}{m\epsilon^2} (\boldsymbol{L}_k \boldsymbol{\Sigma}_k)^\top (\boldsymbol{L}_k \boldsymbol{\Sigma}_k) \right)^{-1} \boldsymbol{R}_k^\top \tag{14}$$

$$= \sum_{k=1}^{K} \boldsymbol{U}_k \boldsymbol{L}_k \boldsymbol{\Sigma}_k \left( \mathbf{I}_m + \frac{1}{\epsilon^2} \boldsymbol{\Sigma}_k^\top \boldsymbol{L}_k^\top \boldsymbol{L}_k \boldsymbol{\Sigma}_k \right)^{-1} \boldsymbol{R}_k^\top \tag{15}$$

$$= \sum_{k=1}^{K} \boldsymbol{U}_k \boldsymbol{L}_k \boldsymbol{\Sigma}_k \left( \mathbf{I}_m + \frac{1}{\epsilon^2} \boldsymbol{\Sigma}_k^2 \right)^{-1} \boldsymbol{R}_k^\top \tag{16}$$

$$= \sum_{k=1}^{K} \boldsymbol{U}_k \boldsymbol{L}_k \left( \mathbf{I}_m + \frac{1}{\epsilon^2} \boldsymbol{\Sigma}_k^2 \right)^{-1} \boldsymbol{\Sigma}_k \boldsymbol{R}_k^\top \tag{17}$$

$$= \sum_{k=1}^{K} \boldsymbol{U}_k \boldsymbol{L}_k \left( \mathbf{I}_m + \frac{1}{\epsilon^2} \boldsymbol{\Sigma}_k^2 \right)^{-1} \boldsymbol{L}_k^\top \boldsymbol{U}_k^\top \boldsymbol{Z} \tag{18}$$

$$= \sum_{k=1}^{K} \boldsymbol{U}_k \mathcal{F} \left( \boldsymbol{L}_k \boldsymbol{\Sigma}_k^2 \boldsymbol{L}_k^\top \right) \boldsymbol{U}_k^\top \boldsymbol{Z} \tag{19}$$

$$= \sum_{k=1}^{K} \boldsymbol{U}_k \mathcal{F} \left( (\boldsymbol{U}_k^\top \boldsymbol{Z})(\boldsymbol{U}_k^\top \boldsymbol{Z})^\top \right) \boldsymbol{U}_k^\top \boldsymbol{Z}. \tag{20}$$

Note that an arbitrary set of orthogonal representatives can be expressed as the principal directions under an orthogonal rotation and scaling, i.e.,

$$\boldsymbol{U}_k^\top \boldsymbol{Q} = \boldsymbol{P}_k \boldsymbol{L}_k \boldsymbol{\Sigma}_k \boldsymbol{\Lambda}_k = \boldsymbol{P}_k \boldsymbol{U}_k^\top \boldsymbol{Z} \boldsymbol{R}_k \boldsymbol{\Lambda}_k, \tag{21}$$

where $\boldsymbol{P}_k \in \mathcal{O}(p)$ denotes an orthogonal rotation within the $k$-th subspace, and $\boldsymbol{\Lambda}_k$ is a diagonal matrix of size $p$. Generalizing the way to form the representatives in (4) from $\boldsymbol{U}_k^\top \boldsymbol{Q} = \boldsymbol{U}_k^\top \boldsymbol{Z} \boldsymbol{A}_k$ to $\boldsymbol{U}_k^\top \boldsymbol{Q} = \boldsymbol{B}_k \boldsymbol{U}_k^\top \boldsymbol{Z} \boldsymbol{A}_k$, where $\boldsymbol{B}_k \in \mathbb{R}^{d \times d}$, we have:

$$\nabla_{\boldsymbol{Z}} R_c(\boldsymbol{Q} \mid \boldsymbol{U}_{[K]}) \propto \sum_{k=1}^{K} \boldsymbol{U}_k \boldsymbol{B}_k^\top \boldsymbol{U}_k^\top \boldsymbol{Q} \left( \mathbf{I}_m + \frac{p}{m\epsilon^2} (\boldsymbol{U}_k^\top \boldsymbol{Q})^\top (\boldsymbol{U}_k^\top \boldsymbol{Q}) \right)^{-1} \boldsymbol{A}_k^\top. \tag{22}$$

---

[14]That is, $\text{rank} \left( \text{softmax} \left( (\boldsymbol{U}_k^\top \boldsymbol{Z})^\top (\boldsymbol{U}_k \boldsymbol{Q}) \right) \right)$.

Similarly, by substituting $\boldsymbol{U}_k^\top \boldsymbol{Q} = \boldsymbol{P}_k \boldsymbol{L}_k \boldsymbol{\Sigma}_k \boldsymbol{\Lambda}_k$, $\boldsymbol{A}_k = \boldsymbol{R}_k \boldsymbol{\Lambda}_k$ and $\boldsymbol{B}_k = \boldsymbol{P}_k$ into (22), we obtain

$$\sum_{k=1}^{K} \boldsymbol{U}_k \boldsymbol{L}_k \boldsymbol{\Lambda}_k \left( \mathbf{I}_m + \frac{1}{\epsilon^2} \boldsymbol{\Lambda}_k^2 \boldsymbol{\Sigma}_k^2 \right)^{-1} \boldsymbol{\Lambda}_k \boldsymbol{L}_k^\top \boldsymbol{U}_k^\top \boldsymbol{Z}. \tag{23}$$

## C  More experimental results

**Training setup.** We train the CBT models in Table 3 and all models in Table 4 150 epochs with the Lion optimizer [64]. The learning rate is $2.0 \times 10^{-4}$, the weight decay coefficient is $0.05$, and the batch size is 256. We also incorporate a warm-up strategy over the first 20 epochs. For data augmentation, we adopt a rather simple choice: just random cropping and random horizontal flipping. We apply label smoothing with a smoothing coefficient of $0.1$. For fine-tuning, we use the AdamW optimizer [65], a learning rate of $5 \times 10^{-5}$, weight decay of $0.01$, and batch size 64. The settings above are largely inherited from [23].

**Quantitative segmentation properties.** Following prior works [6, 51], we evaluate the zero-shot segmentation performance of ImageNet-1K pretrained models on the PASCAL VOC12 validation set [66]. Specifically, we assess the best-performing attention maps of the [CLS] token, but in a simplified setting that considers only three segmentation targets, instead of fine-grained semantic classes. In Table 5, we find that the hybrid model has consistently better segmentation performance. We also measure the segmentation performance of different layers and report the results in Table 6.

Table 5: **Zero-shot segmentation.** We use the Jaccard similarity, which is also called intersection over union (IoU), to quantify the alignment between the [CLS] token's attention map and the segmentation ground truth. The best-performing results are highlighted in bold.

| Segmentation target | (CRATE+CBT)-S | CRATE-S | CBT-S | ToST-S |
|---|---|---|---|---|
| Foreground | **0.68** | 0.65 | 0.51 | 0.42 |
| Background | **0.78** | 0.76 | 0.72 | 0.75 |
| Boundary | **0.18** | 0.17 | 0.15 | 0.12 |

Table 6: **Zero-shot segmentation across layers.** The top-performing three layers for each model are underlined.

| Layers | L1 | L2 | L3 | L4 | L5 | L6 | L7 | L8 | L9 | L10 | L11 | L12 |
|---|---|---|---|---|---|---|---|---|---|---|---|---|
| (CRATE+CBT)-S | 0.37 | 0.44 | 0.55 | 0.46 | 0.56 | 0.58 | 0.60 | 0.53 | 0.59 | 0.56 | 0.55 | 0.55 |
| CRATE-S | 0.34 | 0.45 | 0.51 | 0.48 | 0.53 | 0.56 | 0.53 | 0.57 | 0.55 | 0.52 | 0.50 | 0.39 |
| CBT-S | 0.39 | 0.41 | 0.43 | 0.42 | 0.37 | 0.40 | 0.38 | 0.33 | 0.35 | 0.34 | 0.36 | 0.33 |

Table 7: **Throughput comparisons.** The results below are obtained from experiments on dataset CIFAR-10 with an image resolution of $512 \times 512$.

| Images / sec | CBT-T | ViT-T | CRATE-T | ToST-T | CBT-S | ViT-S | CRATE-S | ToST-S |
|---|---|---|---|---|---|---|---|---|
| Training | **336** | 174 | 203 | 323 | **205** | 94 | 111 | 199 |
| Inference | **572** | 370 | 395 | 533 | **405** | 211 | 220 | 429 |

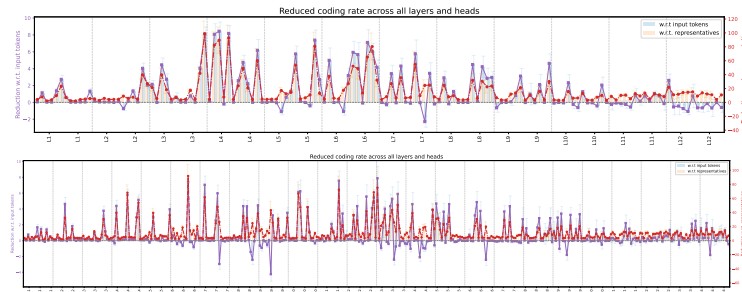

Figure 10: Additional results on CBT-B and CBT-L for the experiment in Fig. 6.

# D  PyTorch implementation

---

**Algorithm 1:** PyTorch implementation of CBSA (8)

---

```python
class CBSA(nn.Module):
    def __init__(self, dim, heads, dim_head):
        super().__init__()
        inner_dim = heads * dim_head
        self.heads = heads
        self.dim_head = dim_head
        self.scale = dim_head ** -0.5
        self.attend = nn.Softmax(dim=-1)
        self.pool = nn.AdaptiveAvgPool2d(output_size=(8, 8))
        # subspace bases
        self.proj = nn.Linear(dim, inner_dim, bias=False)
        # over-parameterization
        self.to_out = nn.Linear(inner_dim, dim)
        # step-sizes
        self.ss_x = nn.Parameter(torch.randn(heads, 1, 1))
        self.ss_rep = nn.Parameter(torch.randn(heads, 1, 1))

    def attention(self, query, key, value):
        dots = (query @ key.transpose(-1, -2)) * self.scale
        attn = self.attend(dots)
        out = attn @ value
        return out, attn

    def forward(self, x):
        b, n, c = x.shape
        height = width = int(n ** 0.5)
        # projection onto subspaces
        w = self.proj(x)
        # representative initialization
        rep = self.pool(w[:, :-1, :].reshape(b, height, width, c).
    permute(0, 3, 1, 2)).reshape(b, c, -1).permute(0, 2, 1)
        w = w.reshape(b, n, self.heads, self.dim_head).permute(0, 2,
    1, 3)
        rep = rep.reshape(b, 64, self.heads, self.dim_head).permute(0,
     2, 1, 3)
        # representative extraction
        rep_delta, attn = self.attention(rep, w, w)
        rep = rep + self.ss_rep * rep_delta
        # representative contraction
        x_delta, _ = self.attention(rep, rep, rep)
        # contraction broadcast
        x_delta = attn.transpose(-1, -2) @ x_delta
        x_delta = self.ss_x * x_delta
        x_delta = rearrange(x_delta, 'b h n k -> b n (h k)')
        # projection back to the ambient space
        return self.to_out(x_delta)
```

---

# E  Limitations

Our work aims to propose a unified framework for both interpretable and efficient attention. Currently, we have proposed a unified optimization objective to derive such an attention mechanism, CBSA, which is capable to accommodate previous interpretable attention mechanisms derived from MCR$^2$. Nonetheless, more fundamental and mathematical connections between our CBSA and existing efficient attention mechanism remain unexplored.

