# OpenReview forum: "Towards Interpretable and Efficient Attention: Compressing All by Contracting a Few"
_NeurIPS.cc/2025/Conference — NeurIPS 2025 spotlight_

### Official Review · Reviewer_PryS · 2025-06-26

**Clarity:** 2
**Significance:** 3
**Originality:** 2
**Rating:** 5
**Confidence:** 3

**Summary:**

The paper proposes a new approach to the computation of the Attention mechanism, called CBSA, that runs in linear time. This approach comes from optimizing an information-theoretic objective, inferred from the maximal coding rate reduction principle. The basic idea is to choose some representatives of the input tokens and contract them to reduce redundancy, then broadcast them and use them to update the full token set.
The CSBA operator proposed is interpretable, it indeed comes from information theory and aims to project over some small dimensional orthonormal basis, it is efficient as it runs in O(n) and it also has a very strong theoretical backbone by being obtained from unrolling an algorithm via GD steps.
Extensive experiments on image classification and segmentation tasks highlight CBSA to be effective and computationally efficient, outperforming channel attention and matching full attention in performance, but with smaller computations required.

**Questions:**

none

**Ethical Concerns:**

["NO or VERY MINOR ethics concerns only"]

**Final Justification:**

Authors have thoroughly addressed my concerns and the paper has improved in clarity. My final score is 5.

**Limitations:**

•	The authors suggest representative selection can be made via SVD, fixing or learning, but they do not provide comparisons among them
•	There is no analysis on the step size specifying how results vary according to them

**Quality:**

4

**Strengths And Weaknesses:**

STRENGTHS
•	The paper provides a rigorous derivation of the algorithm starting from a theoretically well-grounded objective
•	Finds a way to unify two very promising methods like channel and full attention, combining them into a very powerful unified algorithm
•	The procedure is interpretable from an information point of view
•	The algorithm is efficient since it runs linearly in the sequence length
•	The theoretical properties are backed by the empirical results
WEAKNESSES
•	The representative tokens might have undesired behaviors depending on the way they are computed, and this hasn’t been explored
•	No code released yet to see the implementation

---

> ### Author Rebuttal · Authors · 2025-07-29
>
> We sincerely appreciate your recognition of our work and the insightful comments.
>
> > **Weakness 1**: The representative tokens might have undesired behaviors depending on the way they are computed, and this hasn’t been explored.
>
> We are particularly pleased to have the opportunity to address this farsighted concern.
>
> Our findings actually attribute the differences between attention mechanisms to the distinct choices of representatives,
> as **their number and structure** (e.g. orthogonality) induce divergent information propagation patterns (i.e., the behaviors).
> Generally, three cases of behaviors can be analyzed:
> + **Behavior 1**: interactions among the representatives;
> + **Behavior 2**: interactions between input tokens and their representatives;
> + **Behavior 3**: interactions among input tokens.
>
> Undesirable/Desirable behaviors of representatives has been reported in our Section 3.3:
> 1. **Undesired Behavior 1**: orthogonal representatives **do not interact with each other**, resulting the channel-mixer variant Eq. (19);
> 2. **Undesired Behaviors 1 and 2**: Orthogonal and fixed (i.e., input-agnostic) representatives **scale in-place along their fixed directions**, leading to the channel attention Eq. (20).
> 3. **Desired Behavior 3**: In contrast, when input tokens are representatives themselves, their behaviors become **highly flexible and expressive** at the cost of efficiency, deriving the full attention variant Eq. (17).
>
> As the analyses above remain special cases, we propose three promising perspectives to examine their general behaviors:
> 1. **Graph perspective.** We can view **Behaviors 1–3** as the edges (i.e., the information flow pathways) in a graph where nodes are input tokens or their representatives. The more densely connected the graph is, which is determined by the number and structure of the representatives, the more expressive its information propagation becomes, but the harder it is to maintain efficiently.
> When input tokens are themselves representatives, it is a  fully-connected digraph.
> When representatives are orthogonal, it reduces to a bipartite graph.
>
> 2. **Dictionary learning perspective.** We can view the representatives as atoms of a dictionary and recognize that the special cases we studies in Section 3.3 resembles the advances in dictionary learning.
> For example, when the representatives are orthogonal and fixed, they form a complete dictionary.
> When they are orthogonal but input-dependent, the structure resembles a submatrix of an overcomplete dictionary in compressed sensing [80].
> When input tokens are themselves representatives, it becomes a self-expressive dictionary [44].
> Therefore, we can interpret the way the representatives are computed as a form of dictionary learning, and their behaviors as the structure of the atoms in the learned dictionary.
>
> 3. **Rank perspective.**
> We can measure **Behavior 2** via the rank of the broadcast matrix (i.e., the attention matrix computed during the extraction step), which can be interpreted as encoding the edge weights between input tokens and their representatives.
> A significantly low rank suggests the presence of an undesired behavior where many nodes of the representatives become “dead” (i.e., failing to relay information).
> For **Behaviors 1 and 3**, recent studies [62, 63] have shown that the effectiveness of softmax attention can be largely attributed to the attention matrix being consistently full-rank.
>
> **Leveraging the Rank Perspective to Identify Undesired Behavior and Improve CBSA Performance**
> From the rank perspective discussed above,
> we found that the rank of the attention matrix of the extraction (cross-attention) step, i.e,
> $\text{rank} \left( \text{softmax} \left( Z^\top Q_0 \right) \right)$,
> was overly small.
> This suggests that fewer representatives than expected are effectively utilized to gather, process, and broadcast information from the input tokens, which is an **undesired Behavior 2**.
> This low-rank bottleneck stemmed from the learnable initialization of $Q$, and resulted in the unsatisfactory performance of CBSA.
> Now, we adopt a **pooling**-based initialization strategy, i.e., $Q_0 = \text{AvgPool}(Z)$.
> This enables the attention matrix to achieve maximal rank, thereby improving model performance.
> A detailed analysis will be presented in the full paper.
>
> **Revised Table 2**: Fair comparisons between CBSA, TSSA, and MSSA on ImageNet-1K.
> | Models | CBT-T | CBT-S | ToST-T | ToST-S | CRATE-T | CRATE-S |
> | :- | - | - | - | - | - |-|
> | acc@1 (old)  | 62.9| 69.5 | - | - | - | - |
> | acc@1  | **63.2**  | **71.4** |  61.2 | 68.5| 64.7| 72.1|
>
> The performance gain between small size models has been increased from 1% acc@1 to about 3%.
> Please refer to our response to Reviewer FLEn for additional empirical evidences demonstrating the improvements achieved by addressing the undesirable interactions between input tokens and their representatives.
>
> > **Weakness 2**: No code released yet to see the implementation.
>
> Thank you for your interest in the implementation details.
>
> The PyTorch implementation of our CBSA was provided in Appendix: Algorithm 1.
> We will open-source our code and demo upon publication.
> In fact, our implementation builds upon the official PyTorch code from [27], with key modifications to the attention layer.
>
> > **Limitation**: The authors suggest representative selection can be made via SVD, fixing or learning, but they do not provide comparisons among them. There is no analysis on the step size specifying how results vary according to them.
>
> The points raised by the reviewer are highly valuable, as they may further enhance the model’s interpretability and offer deeper insights into its behavior. For instance, one intriguing question is whether the cross-attention mechanism implicitly implements or approximates an SVD-like procedure.
> Another question is to analyze whether adaptive step sizes outperform fixed ones. It is also worth exploring whether the step sizes across different layers or attention heads correlate with their functional importance.
>
> While these questions are highly valuable for better understanding the learning dynamics, a thorough investigation is beyond the scope of this work. We appreciate the reviewer’s feedback and will consider them as promising directions for future research.
>
> We hope the discussions above could resolve your concerns on our work. Please let us know if any further clarification is needed.
>
> [80] Emmanuel J Candes and Terence Tao. "Decoding by linear programming." IEEE Transactions on Information Theory 2005
>
> References [1–61] are provided in the manuscript, references [62–64, 79] are listed in our response to Reviewer FLEn, and references [65-75] are listed in our response to Reviewer  2uTu.

---

> > ### Comment · Reviewer_PryS · 2025-08-07
> >
> > this is a clear accept for me and the rebuttal from the authors only strenghtened the rating i previously gave.

---

### Official Review · Reviewer_AnYY · 2025-07-02

**Clarity:** 3
**Significance:** 3
**Originality:** 3
**Rating:** 5
**Confidence:** 4

**Summary:**

The paper introduces Contract‑and‑Broadcast Self‑Attention (CBSA), a new attention operator derived by unrolling a unified "compress‑all‑by‑contracting‑a‑few" objective. A small set of learnable representative tokens are first extracted by a cross‑attention‑like step, then contracted (self‑attention among the representatives) and finally broadcast back to all tokens via the learned coefficients. With a fixed number of representatives m, CBSA scales linearly in sequence length and subsumes several known mechanisms: it reduces to full self‑attention when every token is its own representative, to an input‑dependent channel mixer when the representatives are orthogonal, and to TSSA‑style channel attention when they are orthogonal & fixed.

Experiments on ImageNet‑1k classification and ADE20K segmentation show that CBSA matches or exceeds the accuracy of MSSA/CRATE while computing 2–3× fewer pair‑wise similarities and retaining desirable interpretability traits such as emergent segmentation and robustness to weight perturbations.

**Questions:**

- Could the authors provide some visualizations of segmentations produced by different models? For example, results on ADE20K.

- Why is “Robustness under parameter perturbation” a desirable property? Would this property be useful in certain practical settings?

- Could the authors provide a detailed mathematical description of the proposed Contract-and-Broadcast Self-Attention (CBSA) in the appendix (i.e., a mathematical version of Algorithm 1: PyTorch implementation of a CBSA layer in the appendix)? This could help readers quickly understand the model architecture.

**Ethical Concerns:**

["NO or VERY MINOR ethics concerns only"]

**Final Justification:**

Most of my questions and concerns have been addressed. Overall, I believe this submission makes interesting and valuable contributions to interpretable and efficient attention mechanisms. I would like to maintain my original score and vote for acceptance.

**Quality:**

3

**Strengths And Weaknesses:**

**Strengths**

- Unified Principle: Provides a single, well‑motivated objective linking interpretability (subspace compression) and efficiency, rather than ad‑hoc architectural tweaks.

- Shows, in closed form, how full spatial attention, channel mixers, and pure channel attention are special cases of CBSA. This is intellectually appealing and may guide future designs.

- For fixed $m$, compute and memory are $O(N)$, yet the operator retains spatial interactions, also with detailed complexity analysis (Table 1) and wall‑clock measurements.

- Competitive Top‑1 on ImageNet and mIoU on ADE20K with fewer pairwise similarity computations; robustness and segmentation emergence are analysed with convincing visualizations.


**Weaknesses**

- [minor] Statistical Rigor: Only some results include error bars (Fig. 2); other tables (e.g., Tables 2–4) lack variance estimates.

- [minor] Writing density – The derivations (Sec 3) are terse; readers unfamiliar with previous work may struggle. Some notation (e.g., gradient signs, scaling constants) could be clarified.

---

> ### Author Rebuttal · Authors · 2025-07-29
>
> We sincerely appreciate your recognition of our work and the constructive suggestions.
>
> > **Weakness 1**: Statistical Rigor: Only some results include error bars (Fig. 2); other tables (e.g., Tables 2–4) lack variance estimates.
>
> As suggested, we will present all results in Figure 2 with their error bars.
>
> However, due to limited resources, we are currently unable to estimate the variance in model performance.
> **We humbly suppose that this concern stems from the marginal performance improvement when comparing our CBSA to TSSA.** If so, we believe our new experimental results will address this.
>
> Our CBSA now substantially outperforms TSSA by simply adopting a pooling-based initialization strategy for the representatives.
> Previously, the representatives are initialized via learnable parameters, which turns out to be the performance bottleneck of our method. Please refer to our response to the **Weakness 1** identified by Reviewer FLEn for more details and improved results.
>
> **Revised Table 2**: Fair comparisons between CBSA, TSSA, and MSSA on ImageNet-1K. The representation compactness is measured in the final attention layer.
>
> | Models | CRATE-T  | CBT-T | ToST-T  | CRATE-S | CBT-S   | ToST-S  |
> | - | - | - | - | -| - | -|
> | $R_c(\boldsymbol{Z}^L \mid \boldsymbol{U}^L_{[K]})$ | 447$\pm$55 | **622**$\pm$**44** | 665$\pm$31 | 987 $\pm$137 | **1317**$\pm$**82** | 1558 $\pm$51 |
> |acc@1| 64.7| **63.2**| 61.2| 72.1 | **71.4** | 68.5|
>
> Results of our method are highlighted **in bold**. The standard deviations are computed over 1,000 images.
>
> We also observe that models with more compact final representations (i.e., smaller $\small R_c(\boldsymbol{Z}^L \mid \boldsymbol{U}^L_{[K]})$) demonstrate superior performance.
> This suggests that $\small R_c(\boldsymbol{Z}^L \mid \boldsymbol{U}^L_{[K]})$ may serve as **a more reliable evaluation metric**.
>
> > **Weakness 2**: Writing density – The derivations (Sec 3) are terse; readers unfamiliar with previous work may struggle. Some notation (e.g., gradient signs, scaling constants) could be clarified.
>
> Thank you for this valuable feedback.
>
> We will continue refining our manuscript to achieve better balance between derivations, conclusions and discussions.
>
> > **Question 1**: Could the authors provide some visualizations of segmentations produced by different models? For example, results on ADE20K.
>
> We will visualize and compare the segmentation masks predicted by different models in the appendix of our final version.
>
> In addition to these qualitative results,
> we report that the best performance of our method on ADE20K has been improved **from 52.0 to 53.3**,
> owing to the modified representative initialization strategy.
> We respectfully refer the reviewer to our response to Reviewer FLEn.
>
> > **Question 2**: Why is “Robustness under parameter perturbation” a desirable property? Would this property be useful in certain practical settings?
>
> **The robustness is desirable because the following aspects**:
> + **It aligns with the derivation**.
> Our CBT theoretically models the attention heads as subspaces, and the projection vectors as basis vectors.
> Therefore, perturbations to these basis vectors do not significantly alter their spanned subspaces, thereby largely preserving the model peformance.
>
> + **It calls for advanced interpretation**.
> The seminal work [76] has already noticed that it is the spanned subspaces rather than the individual units that contains the semantic information in the deeper layers of neural networks. This also aligns with more recent interpretability advances where individual neurons are difficult to interpret [77] and interpretable attention heads can be identified [78].
>
> + **It enables potential applications** in model quantization and severe industrial environments where perfect parameter stability cannot be guanranteed. However, as these fields fall beyond our expertise, these applications remain speculative.
>
> > **Question 3**: Could the authors provide a detailed mathematical description of the proposed Contract-and-Broadcast Self-Attention (CBSA) in the appendix (i.e., a mathematical version of Algorithm 1: PyTorch implementation of a CBSA layer in the appendix)? This could help readers quickly understand the model architecture.
>
> Thank you for this valuable feedback.
>
> We present the mathematical version of Algorithm 1 as follows.
>
> ***
> **Algorithm 1: Contract-and-Broadcast Self-Attention**
> ***
> **Input**: input tokens $Z \in \mathbb{R}^{d \times N}$, number of representatives $m$
>
> **Parameters**:  subspace bases $U_{[K]} = \lbrace U_k \in \mathbb{R}^{d \times p} \rbrace_{k=1}^K$, initialization of representatives $Q_0 \in \mathbb{R}^{d \times m}$ (optional), step-size for input tokens $\kappa$, step-size for representatives $\eta$
>
> **parallel for** $k \in \lbrace 1, \ldots, K \rbrace$ **do**
>
> $\ \ \ \ $\# project input tokens onto the $k$-th subspace
>
> $\ \ \ \ Z_k \coloneqq U_k^\top Z$ $\ \ $\# token projections onto the $k$-th subspace, $Z_k \in \mathbb{R}^{p \times N}$
>
> $\ \ \ \ $\# Initialize the representatives of $Z_k$
>
> $\ \ \ \ $**If** $Q_0$ is parameterized **then**  $\ \ $\# not recommended
>
> $\ \ \ \ \ \ \ \ Q_k \coloneqq U_k^\top Q_0$
>
> $\ \ \ \ $**else** $\ \ $\# recommended
>
> $\ \ \ \ \ \ \ \ Q_k \coloneqq  \text{AvgPool}(Z_k, m)$ $\ \ $\# $Q_k \in \mathbb{R}^{p \times m}$, $\text{AvgPool}$ denotes average pooling
>
> $\ \ \ \ $\# Extract the representatives of $Z_k$
>
> $\ \ \ \  \overline{A}_k \coloneqq \text{softmax}\left(Z_k^\top Q_k \right)$ $\ \ $\# the attention matrix $ \overline{A}_k \in \mathbb{R}^{N \times m}$
>
> $\ \ \ \  Q_k \coloneqq Q_k + \eta Z_k \overline{A}_k$  $\ \ $\# the representatives of $Z_k$
>
> $\ \ \ \ $\# Contract the representatives
>
> $\ \ \ \  Q_k \coloneqq Q_k \text{softmax}\left(Q_k^\top Q_k \right)$ $\ \ $\# the contractions of the representatives
>
> $\ \ \ \ $\# Broadcast the contractions back to input tokens
>
> $\ \ \ \ Z_k \coloneqq \kappa Q_k \overline{A}_k^\top$
>
> **end parallel**
>
> **return** $\sum_{k=1}^{K} U_k Z_k$
> ***
>
> We hope the discussions above could resolve your concerns on our work. Please let us know if any further clarification is needed.
>
> [76] Christian Szegedy et al. "Intriguing properties of neural networks." ICLR 2014
>
> [77] Nelson Elhage et al. "Toy models of superposition." 2022
>
> [78] Catherine Olsson et al. "In-context learning and induction heads." 2022
>
> References [1–61] are provided in the manuscript, references [62–64, 79] are listed in our response to Reviewer FLEn, and references [65-75] are listed in our response to Reviewer  2uTu.

---

> > ### Comment · Reviewer_AnYY · 2025-08-05
> >
> > I would like to thank the authors for providing a detailed rebuttal to my questions. Most of my questions and concerns have been clarified. In particular, I believe that 'Algorithm 1: Contract-and-Broadcast Self-Attention' would be very helpful for readers.

---

### Official Review · Reviewer_2uTu · 2025-07-03

**Clarity:** 3
**Significance:** 3
**Originality:** 3
**Rating:** 5
**Confidence:** 1

**Summary:**

This paper introduces Contract-and-Broadcast Self-Attention (CBSA), a novel attention mechanism designed to jointly address interpretability and efficiency in Transformers. CBSA is derived from a unified optimization objective inspired by Maximal Coding Rate Reduction, where a small set of representative tokens is contracted via self-attention and their effects are broadcast back to the full input. The method is interpretable by construction and scales linearly with input size. The authors demonstrate that CBSA generalizes several known attention mechanisms (e.g., full attention, channel attention) as special cases. Experiments on standard image classification and segmentation tasks show that CBSA performs competitively with reduced computational cost and improved robustness.

**Questions:**

1. It is not entirely clear to me how the initial guess Q0 is optimized during training. While the update rule for Q (e.g., Eq. 10–11) is continuous, Q is supposed to represent a small subset of the input tokens Z, which feels more like a discrete selection. Could the authors clarify how this representative set is extracted and optimized? Are there constraints or assumptions that guide its learning?

2. The proposed CBSA approach seems conceptually similar to LoRA, in that both reduce the representation to a few low-rank basis components. Could the authors explain how CBSA differs from or improves upon LoRA?

3. Does the CBSA framework improve efficiency during inference, or is it primarily beneficial during training? Specifically, do the additional steps—such as projecting into subspaces and broadcasting from representatives—introduce overhead at test time?

**Ethical Concerns:**

["NO or VERY MINOR ethics concerns only"]

**Final Justification:**

The authors provide a thorough response to all of my questions and concerns. Most of them have been addressed with improved clarity.

**Limitations:**

There isn't a limitation section in the paper.

**Quality:**

3

**Strengths And Weaknesses:**

Strengths:
1. The paper introduces a principled objective that unifies interpretability and efficiency, which are typically treated separately in prior attention models.

2. The proposed Contract-and-Broadcast Self-Attention (CBSA) is derived directly from a gradient-based optimization on a compression objective (MCR²). This provides a clear, interpretable foundation for the mechanism.

3. The authors show that MSSA (full attention) and TSSA (channel attention) are special cases of CBSA, depending on the choice of representative tokens. This highlights CBSA as a general and flexible framework.

Weaknesses:
1. CBSA depends on projecting data into orthogonal low-dimensional subspaces, which assumes that meaningful structure lies there. This assumption may not hold for all modalities or tasks, potentially limiting generality.

2. In the experiments, there is no comparison to other methods (e.g., MSSA, TSSA) to show whether CBSA compresses more effectively or more meaningfully. Without a baseline, it’s unclear whether the observed compression is a distinctive advantage or just typical of deep models.

---

> ### Author Rebuttal · Authors · 2025-07-28
>
> Thank you for your insightful comments. It is our pleasure to discuss these points.
>
> > **Weakness 1**: CBSA depends on orthogonal low-dimensional subspaces, which assumes that meaningful structure lies there. This assumption may not hold for all modalities or tasks, potentially limiting generality.
>
> We acknowledge that the structure of **U**nion **o**f **O**rthogonal **S**ubspaces (**UoOS**) faces several challenges in achieving generality, including its adaptation to all modalities and tasks as noted by the reviewer.
> In light of this theoretical gap, we excluded natural language processing tasks from our experiments.
>
> However, we contend that adopting the structure of UoOS is not prohibitively restrictive in practice, though specific theoretical analyses in other modalities and tasks remain future work.
> Underpinned by the following facts:
> + **subspaces are over-parameterized** as each attention layer parameterizes a distinct $U_{[K]}$, yielding $L$ sets of $K$ orthogonal subspaces;
> + tokens can be compressed towards **finer-grained low-dimensional structures** within each of the subspaces,
>
> UoOS could be a feasible starting point or prototype towards the optimal structures, e.g., simplex ETF [65] or its generalized case [66].
>
> Regarding the language modality, an input sentence or article can be projected onto one or more subspaces that acts as its principal subspace  or multiple principal subspaces in the sense of PCA or GPCA [67].
> This process mirrors Latent Semantic Analysis (**LSA**), which is a classical NLP technique.
> Similarly, the channel-mixer variant of our CBSA employs the principal directions as the representatives (see Eqs. (18)-(19)).
> **Therefore, the representatives introduced in our work constitute finer-grained structures than UoOS, dynamically encoding a broader spectrum of semantic concepts.**
>
> > **Weakness 2**: In the experiments, there is no comparison to other methods (e.g., MSSA, TSSA) to show whether CBSA compresses more effectively or more meaningfully. Without a baseline, it’s unclear whether the observed compression is a distinctive advantage or just typical of deep models.
>
> We thank the reviewer for this constructive suggestion.
>
> **The Prevalence of Compression**
>
> Compression is not a phenomenon unique to Transformers built upon the MCR$^2$ objective (e.g., CRATE, ToST, and our CBT), but rather pervasive across deep learning models.
> In a collection of network architectures for classification, their representations are progressively compressed as the layers go deeper and the training continues [68], and eventually converges to neural collapse at the terminal phase of training [65].
> Meanwhile, compression toward low-dimensional distributions can also be observed in generative tasks, e.g., in diffusion models [69].
> And another research direction assesses model performance based on representation compactness [70, 71].
>
> **Compactness Comparison to Other Methods**
>
> We evaluate representation compactness in the final attention layer across related methods, quantified by the compression term $R_c$ of Eq. (2).
>
> We observe that our CBSA compresses more effectively than TSSA but is inferior to MSSA.
>
> More importantly, we find that the final representation compactness reflects the model performance well, indicating that the coding rate can act as a novel evaluation metric.
>
> **Table 8 (new)**: Measuring representation compactness in the final attention layer.
>
> |Models|CRATE-T| CBT-T|ToST-T|CRATE-S|CBT-S|ToST-S|
> |-|-|-|-|-|-|-|
> |$R_c(\boldsymbol{Z}^L \mid \boldsymbol{U}^L_{[K]})$ |447|**622**|665|987|**1317**|1558|
> | acc@1|64.7|**63.2** |61.2|72.1|**71.4** |68.5|
>
> Results of our method are highlighted **in bold**.
>
> Note that the performance results of CBSA have been updated by changing the representative initialization strategy
> (please refer to our response to the weakness 1 identified by Reviewer FLEn for further details).
>
> > **Question 1**: It is not entirely clear to me how the initial guess Q0 is optimized during training. While the update rule for Q (e.g., Eq. 10–11) is continuous, Q is supposed to represent a small subset of the input tokens Z, which feels more like a discrete selection. Could the authors clarify how this representative set is extracted and optimized? Are there constraints or assumptions that guide its learning?
>
> We appreciate the opportunity to clarify this confusing point.
>
> **The representatives are extracted through a continuous process.** The representatives are extracted via cross-attention, where the query is the initial guess and the key and value are the input tokens, i.e., $Q \doteq Q_0 + \eta Z\text{softmax}(Z^\top Q_0)$. This cross-attention can be viewed as a gradient step towards solving the $K$-means centroids of the input tokens. Therefore, the extraction is continuous rather than discrete.
>
> **The extraction can be easily adapted as discrete selection.**
> If we replace the softmax with argmax and remove the residual connection, the extraction becomes
> $Q \doteq Z\text{argmax}(Z^\top Q_0)$.
> This selects the $1$-nearest neighbors of the initial guess from the input tokens, resulting in a discrete selection.
>
> **Assumptions and constraints behind the extraction:**
> + **Assumption 1** (most important): When $Q$ is contracted, $Z$ can be correspondingly compressed (**compressing all by contracting a few**);
> + **Assumption 2**: The $Q$ extracted by cross-attention satisfies Assumption 1 (validated in Figure 2 (right)).
> + **Constraint**: The sole constraint is the model architecture, i.e., the optimization objective underlying its forward pass.
>
> > **Question 2**: Could the authors explain how CBSA differs from or improves upon LoRA?
>
> **A Quick Review of LoRA and CBSA**
>
> LoRA approximates the full fine-tuning update of a pre-trained weight matrix, $\Delta W \in \mathbb{R}^{d\times d}$, via the multiplication of two distinct low-rank matrices, i.e., $\Delta W \approx W_a W_b^\top$, where $W_a, W_b \in \mathbb{R}^{d \times r}$ and $r < d$.
>
> Our CBSA approximates the full attention matrix $A_{\text{full}}  \in \mathbb{R}^{N \times N}$ by the Gram matrix of a low-rank matrix $\overline{A} \in \mathbb{R}^{N \times m}$, i.e., $A_{\text{full}} \approx \overline{A}\ \overline{A}^\top$, where $m <  N$ is number of representatives, and $\overline{A}$ is the attention matrix computed by the extraction (cross-attention) step.
>
> **Similarities and Differences between LoRA and CBSA**
> + **Approximation targets**: LoRA approximates the weight matrix update ($\Delta W$) while CBSA approximates the full attention matrix ($A_{\text{full}}$);
> + **Approximation flexibility**: Both allow tunable rank ($r$ and $m$), unlike linear attention (fixed rank $p$);
> + **Number of matrices**: LoRA uses two low-rank matrices ($W_a, W_b$), while CBSA employs only one ($\overline{A}$).
> + **Approximation hardness**: The intrinsic rank of $\Delta{W}$ is typically small [Table 6, 72], but $A_{\text{full}}$ is always full-rank [62]. Consequently, LoRA matches full fine-tuning performance even with small rank r, whereas CBSA never exactly attains identical performance unless in the self-expressive case (Section 3.3);
> + **Acceleration effects**: LoRA reduces fine-tuning overhead but maintains inference costs, while CBSA accelerates both training and inference;
> + **A unified perspective**:
> According to meta-learning interpretations of linear attention [73,74], $W$ corresponds to slow weights (updated only during training), while $A_{\text{full}}$ corresponds to fast weights (input-dependent). Therefore, LoRA approximates the slow-weight update and CBSA directly approximates the fast weight. Both methods are weight low-rank approximation techniques.
>
> > **Question 3**: Does the CBSA framework improve efficiency during inference, or is it primarily beneficial during training? Specifically, do the additional steps—such as projecting into subspaces and broadcasting from representatives—introduce overhead at test time?
>
> We thank the reviewer for this constructive suggestion.
>
> We measure computational throughput in processed images per second (img/s) across Transformer architectures, using a standard $512 \times 512$ resolution typical for semantic segmentation tasks.
>
> |Models|ViT-T|CRATE-T|CBT-T|ToST-T|ViT-S|CRATE-S|CBT-S|ToST-S|
> |-|-|-|-|-|-|-|-|-|
> |im/sec (training)|174|203|**336**|323|94|111.4| **205**|199|
> |im/sec (inference)|370|395|**572**|533|211| 220.3|**405**|429|
>
> The results show these extra steps save more time than they cost.
>
> > **Limitation**: There isn't a limitation section in the paper.
>
> Thank you for this feedback.
>
> While our limitation discussion was originally confined to Appendix D, we will now include a reference to it in the main text to ensure visibility for interested readers.
>
> We hope the discussions above could resolve your concerns on our work. Please let us know if any further clarification is needed.
>
> [65] Vardan Papyan et al. "Prevalence of neural collapse during the terminal phase of deep learning training", PNAS 2020
>
> [66] Jiachen Jiang et al. "Generalized neural collapse for a large number of classes", 2023
>
> [67] Rene Vidal et al. "Generalized principal component analysis (GPCA)", TPAMI 2005
>
> [68] Hangfeng He et al. "A law of data separation in deep learning", PNAS 2023
>
> [69] Peng Wang et al. "Diffusion models learn low-dimensional distributions via subspace clustering", CPAL 2025
>
> [70] Lai Wei et al. "Diff-erank: A novel rank-based metric for evaluating large language models", NeurIPS 2024
>
> [71] Xianzhen Luo et al. "Is Compression Really Linear with Code Intelligence?" 2025
>
> [72] Edward J Hu et al. "Lora: Low-rank adaptation of large language models", ICLR 2022
>
> [73] Jürgen Schmidhuber. "Learning to control fast-weight memories: An alternative to dynamic recurrent networks", Neural Computation 1992
>
> [74] Imanol Schlag et al. "Linear transformers are secretly fast weight programmers", PMLR 2021
>
> [75] Lemeng Wu et al. "Centroid transformers: Learning to abstract with attention", 2021

---

> > ### Comment · Reviewer_2uTu · 2025-08-06
> >
> > Thank you for the thorough response to all of my questions and concerns. Most of them have been addressed with improved clarity. I will raise my score to 5 and encourage the authors to update the manuscript accordingly.

---

### Official Review · Reviewer_FLEn · 2025-07-03

**Clarity:** 3
**Significance:** 3
**Originality:** 2
**Rating:** 5
**Confidence:** 3

**Summary:**

Attention in modern deep learning models has led to significant performance in many domains, but it suffers from a quadratic computational overhead. Additionally, there is still a lack of understanding of their underlying mathematical optimization objective. While prior works work on either of these problems, this paper claims to address both problems with their Contract-and-Broadcast Self-Attention (CBSA). CBSA compresses the input tokens by using a subset of them to create a mixture of low-dimensional subspaces that preserve the original coding rate of the entire input token space. CBSA is a self-attention implementation made interpretable by unfolding a gradient-based optimization on the coding rate preservation objective. Experimental results show competitive performance, computational efficiency, and kept emergent properties such as segmentation and parameter perturbation robustness of their method.

**Questions:**

- Figure 2 (right) is still a bit unclear to me. Is the y-axis suppose to be labeled $\Delta R$ and thus measure Eq. 2 just between $R_c(Z | U_{[K]})$ (blue) and $R_c(Q | U_{[K]})$ (orange)?

**Ethical Concerns:**

["NO or VERY MINOR ethics concerns only"]

**Final Justification:**

Based on my initial review, and the rebuttal on my initial concerns on clarity, marginal performance improvements to related works and the strength of segmentation abilities being resolved, my score is 5.

**Limitations:**

- See weaknesses
- CBSA seems more like an alternative to TSSA than an improved method with regards to performance and computational efficiency.

**Quality:**

3

**Strengths And Weaknesses:**

Paper Strengths:
- The paper provides a novel method for lower the computational complexity of the attention mechanism made interpretable through algorithmic unrolling.
	- Shown in both classification and segmentation tasks.
- The method is linked to existing methods of MSSA if full attention is kept by having the number of representative equal to the number of tokens but also reduces to channel attention as in TSSA if representatives are orthogonal and fixed.
- The model's parameters are significantly more robust than other methods reported.
- The paper is well written and organized.

Paper Weaknesses:
- Other than parameter robustness, this method isn't much more efficient or stronger in performance than TSSA (ToST)
- The claim of segmentation properties is based on one visual analysis in Figure 4 without any quantitative experiments showing it's segmentation performance. While the visual analysis is enough to show sign of segmentation abilities, it's unclear how strong this emergent property is.

---

> ### Author Rebuttal · Authors · 2025-07-27
>
> Thank you for your valuable comments.
>
> To address the weaknesses and limitations above, we present new experimental results demonstrating:
> + **Improved Performance**: With an improved initialization strategy of the representatives, CBSA outperforms TSSA remarkably while maintaining its efficiency.
> + **Quantitative Segmentation**: We measure the segmentation properties using Jaccard similarity and observe that CBSA preserves or even enhances the emergent segmentation properties.
>
> In the following, we will address each of the Weaknesses/Questions/Limitations point-to-point and provide the corresponding experimental results.
>
> > **Weakness 1**: This method isn't much more efficient or stronger in performance than TSSA (ToST).
>
> **Pooling-based initialization breaks through the performance bottleneck.**
>
> Given that CBSA enjoys much more expressiveness than TSSA in principle, it is also confused us for the marginal performance gain. We found that the rank of the attention matrix of the extraction (i.e., the cross-attention) step, i.e,
> $\text{rank} \left( \text{softmax} \left( Z^\top Q_0 \right) \right)$,
> was overly low.
> This suggests that there are fewer representatives being utilized than expected.
> This low-rank bottleneck stemmed from the learnable initialization of $Q$, and resulted in the unsatisfactory performance of CBSA. Now, we adopt a pooling-based initialization strategy, i.e.,
> $Q_0 = \text{AvgPool}(Z)$.
> This enables the attention matrix to achieve maximal rank, thereby improving model performance.
> This intriguing finding aligns with the recent advances in linear attention mechanisms [62,63].
> A detailed analysis will be presented in the final paper.
>
> Consequently, we revise the Tables as follows, highlighting new results **in bold**.
>
> **Revised  Table 1**: Evaluating CBT on ImageNet-1K.
> | Models | CBT-T | CBT-S  | CBT-B  | CBT-L | ViT-S |
> | :- | - | - | - | - | - |
> | FLOPs  | 1.1G         | 4.0G         | 15.1G        | 47.3G        | 9.8G  |
> |acc@1 (old) | 62.9 | 69.5 | 71.7 | 72.9 | - |
> | acc@1 | **63.2**  | **71.4**  | **73.4**  | **74.4**  | 72.4  |
>
> **Revised Table 2**: Fair comparisons between CBSA, TSSA, and MSSA on ImageNet-1K.
> | Models | CBT-T | CBT-S | ToST-T | ToST-S | CRATE-T | CRATE-S |
> | :- | - | - | - | - | - |-|
> | acc@1 (old)  | 62.9| 69.5 | - | - | - | - |
> | acc@1  | **63.2**  | **71.4** |  61.2 | 68.5| 64.7| 72.1|
>
> **Revised Table 3**: Evaluating CBT on ADE20K.
> | Models | CBT-B  | CBT-L  | DEPICT-L [29] | Seg-L-Mask [48] |
> | :- | -| - | - | - |
> | Decoder FLOPs   | 7.8G | 22.3G  | 35.0G | 113.2G |
> | mIoU (old)  | 48.7 | 52.0 | - | - |
> | mIoU   | **49.3** | **53.3** | 52.9 | 51.8 |
>
> **Key improvements**:
> + **Comparable to ViT**: CBT-S performs comparably to ViT-S (71.4 v.s. 72.4) with only 40% of the FLOPs (see revised Table 1);
> + **Superiority to TSSA**: The performance gain between small size models has been increased from 1% acc@1 to about 3% (see revised Table 2);
> + **Better Segmenation**:
>  For semantic segmentation, our CBSA outperforms MSSA and MHSA (see revised Table 3).
>
> **An Additional Experiment: CBSA matches or even exceeds Linear Attention**
>
> To further validate the performance of CBSA, we utilize open-source ViT models pre-trained on ImageNet-21K and modify their forward passes to implement either CBSA or linear attention mechanisms. All models are then fine-tuned on ImageNet-1K.
>
> **Table 7 (new)**: Evaluating pre-trained ViTs adapted to linear complexity on ImageNet-1K.
>
> | Model Sizes | Tiny@224$^2$| Small@224$^2$| Base@224$^2$| Large@224$^2$| Tiny@384$^2$| Small@384$^2$| Base@384$^2$| Large@384$^2$|
> |:-|-|-|-|-|-|-|-|-|
> |MHSA |76.9 | 81.4 | 84.7| 85.9 |79.7| 83.7| 85.9| 87.1|
> |LinearAttn|73.2 | 78.6 | 79.8 | 80.7| 74.9 | 80.1 | 81.8 | 82.2|
> |CBSA |**72.2** | **78.3** | **79.2** | **79.8** | **72.3** | **79.3** | **80.8** | **81.2** |
> |CBSA$^\vee$|**74.7** |**79.9** |**80.9**  |**83.8**| **77.3** | **81.4** | **83.1** |**85.0**|
>
> Pooling-based initialization is adopted.
> $\vee$: the contraction (self-attention) step in CBSA is removed.
> In this case, CBSA effectively derives another existing attention mechanism known as Agent Attention [79].
> A detailed discussion of the practical advantages of removing this step will be provided in the final version of the full paper.
>
> > **Weakness 2**: While the visual analysis is enough to show sign of segmentation abilities, it's unclear how strong this emergent property is.
>
> We thank the reviewer for this constructive suggestion.
>
> We supplement Figure 4 with quantitative experiments evaluating zero-shot segmentation performance.
> Following prior works [8, 47], we assess the [CLS] token attention maps on the validation set of PASCAL VOC12 [64].
> However, as neither [8, 47] provides open-source code, our implementation will differ (demo will be released via a Colab Notebook).
>
> **Jaccard similarity** is also called intersection over union (IoU).
> We adopt this metric to quantify the alignment between the attention map and the segmentation ground truth. Higher Jaccard similarity indicates more pronounced segmentation properties emerged in the model.
>
> **Table 5 (new)**: The Jaccard similarity on the validation set of PASCAL VOC12.
> | Models | (CRATE+CBT)-S | CRATE-S | CBT-S | ToST-S |
> | :- | - | - | - | - |
> | Foreground |0.68 | 0.65 | 0.51  | 0.42 |
> | Background | 0.78 | 0.76 | 0.72 | 0.75 |
> | Boundaries | 0.18| 0.17 | 0.15 | 0.12 |
>
> **Table 6 (new)**: The Jaccard similarity across layers.
> | Layers   | L1   | L2       | L3       | L4       | L5   | L6       | L7       | L8       | L9       | L10      | L11      | L12  |
> | :-| - | - | -| - | - | - | - | - | - | - | - | - |
> | (CRATE+CBT)-S | 0.37 | 0.44     | 0.55     | 0.46     | 0.56 | **0.58** | **0.60** | 0.53     | **0.59** | 0.56     | 0.55     | 0.55 |
> | CRATE-S       | 0.34 | 0.45     | 0.51     | 0.48     | 0.53 | **0.56** | 0.53     | **0.57** | **0.55** | 0.52     | 0.50     | 0.39 |
> | CBT-S         | 0.39 | **0.41** | **0.43** | **0.42** | 0.37 | 0.40     | 0.38     | 0.33     | 0.35     | 0.34     | 0.36     | 0.33 |
>
> The top-performing three layers for each model are highlighted **in bold**.
>
> **Key observations**:
> + The segmentation properties are gradually reduced from MSSA to CBSA to TSSA.
> + The segmentation properties are enhanced in the hybrid model, CRATE+CBT, surpassing pure MSSA.
>
> > **Question**: Figure 2 (right) is still a bit unclear to me. Is the y-axis suppose to be labeled and thus measure Eq. 2 just between (blue) and (orange)?
>
> We sincerely appreciate this feedback and apologize for any confusion caused by the incomplete definition of $\Delta R_c$ in our original submission.
>
> **Definition of $\Delta R_c$.**
> The compression term $R_c$ measures the compactness of input tokens or their representatives.
> We measure the change of $R_c$, i.e., $\Delta R_c$, produced by CBSA to quantify its effect in the compression procedure.
> Note that this is different from the $\Delta R \doteq R(Z) - R_c(Z \mid U_{[K]})$ in Eq.(2).
> We present formal definitions:
> + $\Delta R_c$ w.r.t $Z^\ell$ (**the blue line**) is
> $\Delta R_c \left(Z^\ell  \mid U_{[K]} \right) \doteq R_c\left(\text{CBSA}\left(Z^\ell \mid U_{[K]}\right)  \mid U_{[K]} \right) -  R_c(Z^\ell \mid U_{[K]})$,
> which reflects the extent to which input tokens are compressed by the $\ell$-th CBSA layer;
> + $\Delta R_c$ w.r.t $Q^\ell$ (**the orange line**) is $\Delta R_c(Q^\ell) \doteq R_c\left(\text{MSSA}\left(Q^\ell \mid U_{[K]}\right)\right) -  R_c(Q^\ell)$, where the MSSA over $Q$ corresponds the contraction step of CBSA
> (i.e., the contraction term of Eq. (16)). Therefore, this reflects the extent to which representatives are contracted by the $\ell$-th CBSA layer.
>
> **Figure 2 (Right) Interpretation.** In theory, we expect both the blue and orange lines are consistently above $x=0$, so that the input tokens and their representatives are compressed and contracted in every attention layers. In Figure 2 (Right), we find only 2 of the 12 layers violates this pursuit.
>
>
> > **Limitation**: CBSA seems more like an alternative to TSSA than an improved method with regards to performance and computational efficiency.
>
> Please refer to our response to Weakness 1.
>
> We hope the experiments above could resolve your concerns on our work. Please let us know if any further clarification is needed.
>
> [62] Dongchen Han et al. "Flatten transformer: Vision transformer using focused linear attention." ICCV 2023
>
> [63] Qihang Fan et al. "Breaking the low-rank dilemma of linear attention." CVPR 2025
>
> [64] Mark Everingham et al. "The pascal visual object classes challenge 2012 (voc2012)"
>
> [79] Dongchen Han et al. "Agent attention: On the integration of softmax and linear attention." ECCV 2024.
>
> References [1–61] are provided in the manuscript.

---

> > ### Comment · Reviewer_FLEn · 2025-08-06
> >
> > Thank you for the response to my concerns! My concerns on clarity, marginal performance improvements to related works and the strength of segmentation abilities have been met! I'm increasing my score to 5.

---

### Decision · Program_Chairs · 2025-09-17

**Decision:**

Accept (spotlight)

**Comment:**

This paper proposes Contract-and-Broadcast Self-Attention (CBSA), a unified mechanism addressing Transformer attention’s interpretability and efficiency via a "compress-all-by-contracting-a-few" objective. Reviewers uniformly praised its strengths: a well-motivated unified principle linking subspace compression and efficiency, theoretical derivations showing it subsumes full and channel attention as special cases, linear computational scaling, and competitive performance on ImageNet and ADE20K with fewer computations.
However, concerns were noted: limited variance estimates in results, terse derivations in Section 3, under-explored representative token behaviors, lack of code release, and insufficient quantitative analysis of segmentation properties.
Despite these, the work makes valuable contributions by unifying key attention mechanisms under a rigorous framework with strong empirical support. Its potential to guide future attention design and bridge efficiency-interpretability gaps is significant. Thus, the AC concurs this is a valuable contribution and recommends acceptance.